# Phosphorus-solubilizing bacteria improve the growth of *Nicotiana benthamiana* on lunar regolith simulant by dissociating insoluble inorganic phosphorus

Yitong Xia[1], Yu Yuan [2], Chenxi Li[3] & Zhencai Sun [1✉]

In-situ utilization of lunar soil resources will effectively improve the self-sufficiency of bior-egenerative life support systems for future lunar bases. Therefore, we have explored the microbiological method to transform lunar soil into a substrate for plant cultivation. In this study, five species of phosphorus-solubilizing bacteria are used as test strains, and a 21-day bio-improving experiment with another 24-day *Nicotiana benthamiana* cultivation experiment are carried out on lunar regolith simulant. We have observed that the phosphorus-solublizing bacteria *Bacillus mucilaginosus*, *Bacillus megaterium*, and *Pseudomonas fluorescens* can tolerate the lunar regolith simulant conditions and dissociate the insoluble phosphorus from the regolith simulant. The phosphorus-solubilizing bacteria treatment improves the available phosphorus content of the regolith simulant, promoting the growth of *Nicotiana benthamiana*. Here we demonstrate that the phosphorus-solubilizing bacteria can effectively improve the fertility of lunar regolith simulant, making it a good cultivation substrate for higher plants. The results can lay a technical foundation for plant cultivation based on lunar regolith resources in future lunar bases.

[1] College of Agronomy and Biotechnology, China Agricultural University, Haidian District, Beijing, China. [2] College of Engineering, China Agricultural University, Haidian District, Beijing, China. [3] College of Horticulture, China Agricultural University, Haidian District, Beijing, China. ✉email: zhencai_sun@cau.edu.cn

Series of successful unmanned lunar scientific research programs[1] have aroused worldwide scientific interest in the moon[2]. Considering the value of the moon in future deep space exploration and the exploitation of lunar natural resources[3–6], it is necessary to establish a permanent crewed lunar station. Transporting essential materials (e.g., food and water) for human life support by cargo rockets has become a traditional method in crewed space exploration. However, the tremendous distance between the Earth and the moon makes it uneconomic to use cargo rockets to transport these materials to the moon[7], and even economically unsustainable to provide terrestrial resources to an extraterrestrial permanently crewed station from the perspective of its expected lifespan[8], because of the high cost and resource/energy requirements and the difficulty and time required to plan and execute a launch. A biological regenerative life support system (BLSS) is the only advanced life support system that human beings can rely on to carry out long-term, long-distance, and multi-crewed space missions in the future[9–11]. An efficient BLSS must be able to purify water, produce food, and revitalize the atmosphere in a closed system[12–14], which greatly reduces the Earth-moon transportation frequency, and effectively cut the economic cost of the initial construction and long-term maintenance of the lunar base[15].

Higher plants have the functions of providing food and $O_2$, absorbing $CO_2$, and purifying wastewater. Therefore, they are regarded as the core components of highly closed BLSS[16,17]. However, among current BLSS simulation tests[18–20], the cultivation of higher plants is highly dependent on hydroponic systems, or cultivation substrate based on earth soil[21–25]. Given the vast amount of materials required to construct such systems, it is not economically realistic to carry out these designs in extraterrestrial stations. A natural idea is: why not introduce the unconsolidated fine granular weathering on the lunar surface, or the lunar regolith, to BLSS as a cultivation substrate for higher plants, in order to cut the cost of lunar base construction[26]. The potential of this method, which is known as in-situ resource utilization (ISRU), has been preliminarily studied. Analysis and evaluation have been made on the elemental composition and bioavailability of lunar soil[27] and the ability of the regolith on extraterrestrial planets to support the growth of microorganisms[8,28,29] and plants[30–32]. Studies have shown that the composition of the elements essential for plant growth in lunar regolith is very similar to those in the Earth's soil[27]. Unfortunately, the fertility of lunar regolith is greatly poorer than that of the Earth soil, due to the influence of lunar environmental conditions. It not only lacks carbon and nitrogen nutrients necessary for plant growth but also holds other elements mainly in an insoluble form, which is difficult to be absorbed by plants. Because of this "inertness", the lunar soil cannot provide the absorbable soluble nutrients for plant growth as the Earth's soils do.

An interesting study conducted by Paul et al. [32]. showed that *Arabidopsis thaliana* could be seeded and grow directly on lunar regolith with the support of exogenous nutrient solutions. However, compared with the volcanic rock control, *Arabidopsis thaliana* grown on lunar regolith exhibits slower growth and a severe stress phenotype, which is confirmed by transcriptome data. This indicates that it is necessary to develop some technical measures to improve the physical and chemical properties of lunar regolith and increase the availability of essential plant nutrients before the establishment of a practical extraterrestrial higher plant cultivation system that effectively provides life support in the lunar station.

Reviewing the process of the earth's terrestrial ecosystem evolving hard rocks into porous and biologically active soils, it has been believed that microbial groups (e.g., bacteria and fungi)

are one of the most important factors leading this process[33,34]. Interactions between microbial metabolites and naked rock resulted in the decomposition of silicate, phosphate, carbonate, oxide, and sulfide minerals and the dissolution of some important elements (Si, Al, Fe, Mg, Mn, P, Na, Ti, etc.) from minerals[35–38], which improves the fertility level of regolith and enables the growth of plants. However, the feasibility study on microbial weathering as an improvement measure for extraterrestrial regolith has not been confirmed by academia yet. There are only a few relevant studies, that mainly focus on two directions: Exploration of the ability of cyanobacteria and other autotrophic nitrogen-fixing microorganisms to improve lunar soil fertility[8,39–41], and the very preliminary verification of the improving ability that artificial microorganisms coenosis have on the growth of higher plants in a lunar regolith simulant[30,42]. Unfortunately, considering phosphorus, a major element required by plants and an important component of soil fertility, and stands for about 1% of the total mass in the lunar regolith, current studies either lack experimental evidence that microbial activity directly decomposes mineral structures and leads to phosphorus dissociation[30], or even have obtained contrary experimental evidence[39] (in this study, cyanobacteria activity even led to a decrease in phosphorus content in the simulated lunar soil).

Here, setting phosphorus as the object of the study, five kinds of phosphorus-solubilizing bacteria (PSBs) were introduced to the lunar surface regolith simulant (Fig. 1a). Through detailed experimental design and analysis, a 21-day culture experiment was carried out using a shaking flask method, measured the dynamics of phosphorus content in different forms in the culture medium on a time scale and screened out three strains with the strong phosphorus-solubilizing ability to the lunar regolith simulant. This study verified the potential of soil microorganisms to sustain the growth of plants by dissociating insoluble phosphorus from the lunar regolith, which was again confirmed by phenotypic data in a further *Nicotiana benthamiana* cultivation experiment.

## Results and discussion

**The growth of five PSBs on the lunar regolith simulant.** Considering that there is only a very small amount of lunar regolith material brought back by the Apollo missions and a series of unmanned probes to Earth, lunar regolith simulants or similar minerals are used in place of true lunar regolith[30,39,41,42] in most studies related to the agricultural feasibility of lunar regolith. These simulants, often made from volcanic scoria, used real lunar samples for reference, to achieve good similarity in mineralogy, physicochemical properties, and hydrological properties[43]. The simulant used in our study is a copy of the CAS-1 lunar soil simulant (Methods, Lunar surface regolith simulant materials), which has a very similar elemental composition to Apollo 14 samples (Fig. 1b, c).

Five types of PSBs, namely *Bacillus mucilaginosus*, *Bacillus megaterium*, *Bacillus subtilis*, *Bacillus licheniformis*, and *Pseudomonas fluorescens* were selected for the experiment. Their ability to decompose insoluble inorganic phosphorus was verified in the previous $Ca_3(PO_4)_2$ decomposition test (Fig. 1d). After seven days of culture, the $Ca_3(PO_4)_2$ in the culture medium was decomposed, and the concentration of soluble inorganic phosphorus in the liquid medium was significantly increased by 212.7–519.7% compared with that before culture ($p = 0.001$ or less, $n = 6$ for each PSB). These results indicated that the five PSBs had a strong potential to dissociate inorganic phosphorus elements from $Ca_3(PO_4)_2$. However, whether they can activate inorganic phosphorus in the lunar soil simulants remains to be further verified.

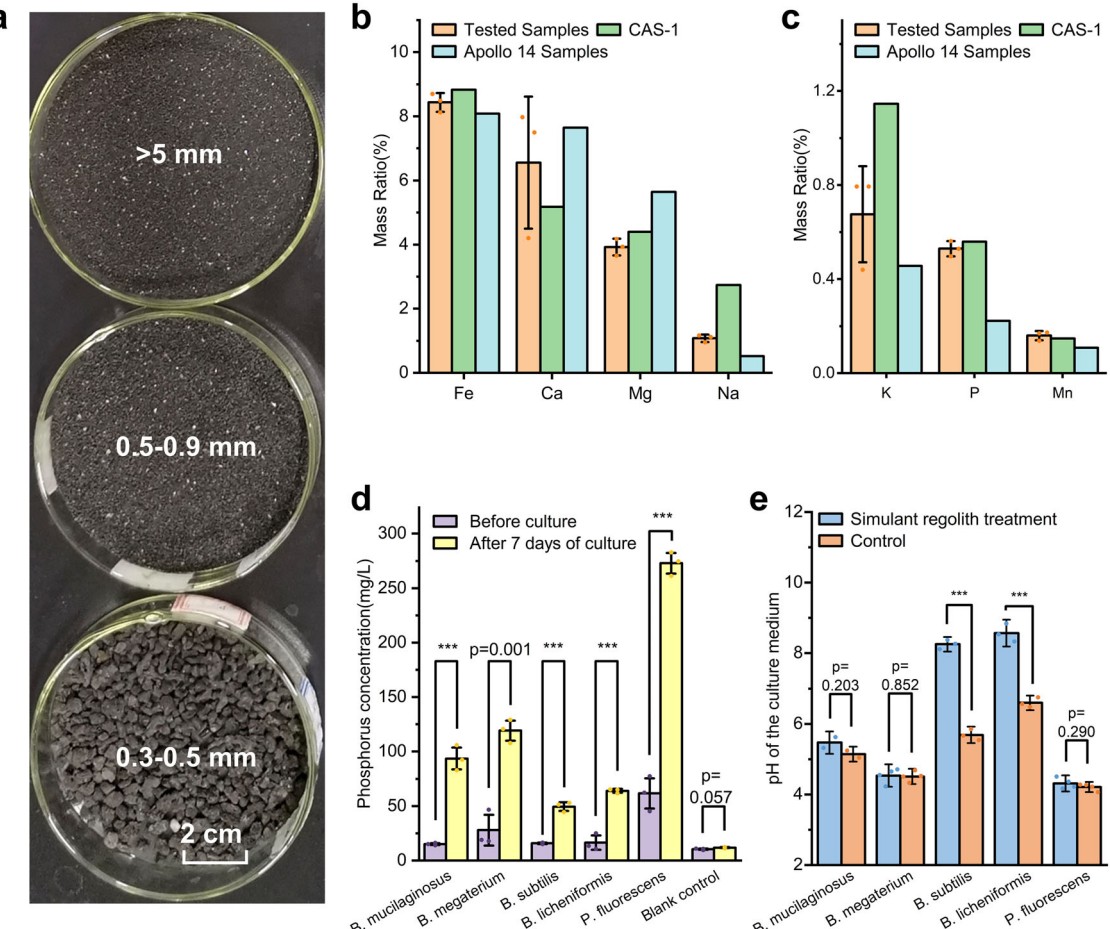

**Fig. 1 Preparations for the culture experiment. a** Photography of lunar surface regolith simulant samples of different particle sizes. The simulants in the plates are >5 mm, 0.5–0.9 mm, and 0.3–0.5 mm from up to down respectively. **b**, **c** The determination of the elemental composition of the tested samples of the regolith simulant and the comparison with CAS-1 lunar regolith simulant and the real lunar regolith samples brought back in the Apollo 14 mission. Data of CAS-1 and Apollo 14 samples that used in this figure is provided by Zheng et al. [80]. All error bars represent Standard Deviation. **d** The identification of all five PSBs' ability to decompose insoluble inorganic phosphorus compounds in the $Ca_3(PO_4)_2$ decomposition test. Different species of inoculated bacteria contain different amounts of phosphorus, which leads to different levels of phosphorus at the beginning of the test. All error bars represent Standard Deviation. *** indicates $p < 0.001$. Significance analyses are conducted using one-side one-way ANOVA tests. **e** The comparison of the pH of the culture medium at 7 DAI between the lunar regolith simulant treatments and their corresponding control in the PSBs culture experiment. All error bars represent Standard Deviation. *** indicates $p < 0.001$. Significance analyses are conducted using one-side one-way ANOVA tests.

The ability to reproduce and grow normally in lunar regolith plays an important role in the function of PSBs to decompose phosphate minerals in lunar regolith particles. Therefore, by measuring the 600 nm light absorbance, or the $OD_{600}$, of the culture at different culture periods, we plotted the growth curve of five PSBs and studied the impacts of lunar regolith simulant on the growth of five PSBs in the shaking flask culture system (Fig. 2a). The mass ratio of regolith simulant and LB medium was 1:2. In general, no matter whether the lunar regolith simulant was added, the five PSBs showed typical growth characteristics, which include an obvious logarithmic growth period followed by a plateau period. This indicates that some components (e.g., peroxides and heavy metals) in the simulant as well as the real lunar regolith that it referred to, do not reveal serious toxicity that inhibits microbial growth.

By comparing the $OD_{600}$ of five PSB treatments in the culture of simulant treatments with the control on a time scale., it was found that the addition of the simulant had no obvious effect on the growth and reproduction of five PSBs in the early stage of culture. The growth curves of the five PSBs almost coincided with their corresponding control, from the inoculation to the pre-

logarithmic and mid-logarithmic stages. The longest coincident relationship could extend to 84 h after the inoculation (HAI) in the *B. subtilis* treatment. This coincident relationship suggests that the simulant has little impact on the growth of PSBs and reproduction at the beginning of culture when it is considered to be "inert". This may be due to the very weak interaction between the simulant particles and the PSBs in this stage. In contrast, from the late logarithmic stage to the plateau stage of the culture, the coincidence of growth curves between the simulant treatment and the corresponding control gradually faded, and the different reactions between the strains were shown. Specifically, some strains had higher $OD_{600}$ in the culture at the plateau stage in simulant treatments than control, such as *B. mucilaginosus* and *B. licheniformis* treatments. On the contrary, the $OD_{600}$ of *B. megaterium*, *B. subtilis*, and *P. fluorescens* was lower than that of the corresponding control, indicating that the later growth of the three PSBs was inhibited by simulants, and this inhibition impact increased with time. This suggests that the plateau period is the key period for the interactions between PSBs and simulants. Therefore, it is necessary to study the dynamic changes of medium composition on a time scale to further clarify the

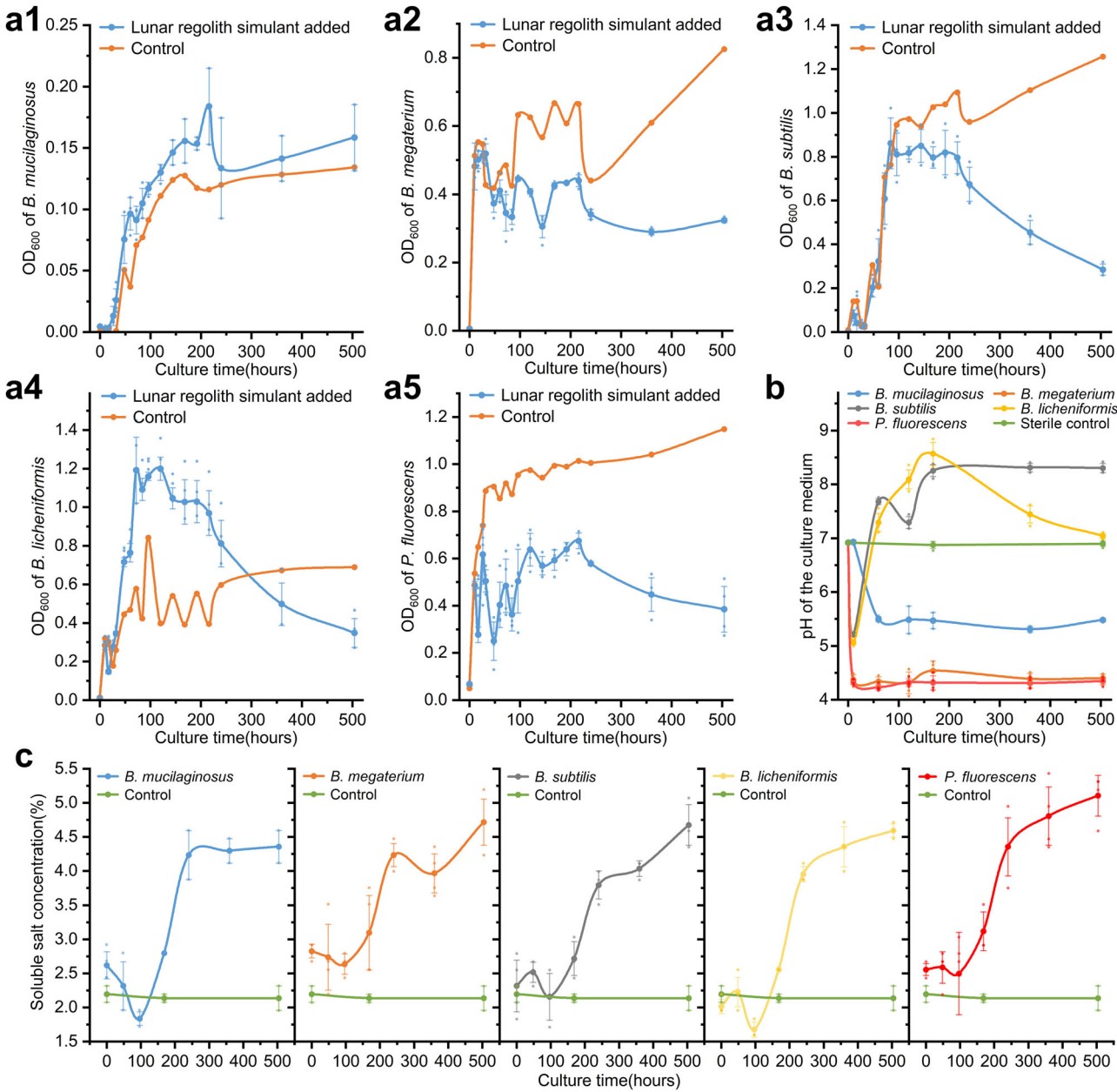

**Fig. 2 Growth curves of five PSBs and the dynamics of pH and the soluble salt content of the culture medium. a1–a5** The growth curves of five PSBs in the regolith simulant treatment and corresponding control during the culture period of 21 days. a1 to a5 represent *B. mucilaginosus, B. megaterium, B. subtilis, B. licheniformis,* and *P. fluorescens*, respectively. The mass ratio of regolith simulant and LB medium was 1:2. All error bars represent Standard Deviation. **b** The dynamics of the pH of the culture medium of five PSB treatments. All error bars represent Standard Deviation. **c** The dynamics of the soluble salt content in the culture medium of five PSB treatments. All error bars represent Standard Deviation.

inhibitory factors and mechanisms that simulants have on the growth of PSBs.

**Influential mechanisms of the lunar regolith simulant on microbial growth.** Two indexes including soluble salt content and pH value of the culture medium were selected to measure liquid samples at different culture times (Fig. 2b, c). All five PSBs induced a rapid increase of soluble salt content at 4–10 days after the inoculation (DAI), which reached 3.80–4.36% at 10 DAI and gently increased to 4.36–5.11% at 21 DAI. However, there was little difference among the five PSBs generally. Considering that there was no significant change ($p = 0.933$, $n = 6$) in the soluble salt content of the control group, we believe that the activity of PSBs led to an increase in soluble salt content. Compared with

soluble salt content, the temporal dynamics of pH in the culture medium of different PSBs showed an obvious difference. The initial pH was determined by the medium itself, which was about 7.0. At 60 HAI, the pH of the culture medium of different strains changed dramatically and reached stability (except for *B. licheniformis*) after 7 DAI. The stable pH values of *B. mucilaginosus, B. megaterium,* and *P. fluorescens* are lower than 7 and are acidic, while the stable pH values of *B. subtilis* and *B. licheniformis* are higher than 7 and are alkaline. The pH of the culture medium in lunar regolith simulant treatments with that of the corresponding control at 7 DAI (Fig. 1e) was further compared. When the simulant was not added, the culture medium of *B. subtilis* and *B. licheniformis* were both weakly acidic, while when the simulant was added, the two PSBs were strongly induced by the simulant,

**Table 1 Results of linear fitting analysis between GIR and soluble salt content and pH of culture medium.**

| PSBs | Linear fitting equation between GIR and salinity | $R^2$ | Linear fitting equation between GIR and pH | $R^2$ |
|---|---|---|---|---|
| *B. mucilaginosus* | $y = 9.1807x - 54.269$ | 0.1212 | $y = 38.434x - 264.56$ | 0.0615 |
| *B. megaterium* | $y = 12.245x - 8.4898$ | 0.3566** | $y = 12.4x - 19.633$ | 0.0096 |
| *B. subtilis* | $y = 20.721x - 29.372$ | 0.6938*** | $y = -1.4785x + 42.075$ | 0.0014** |
| *B. licheniformis* | $y = 31.688x - 124.44$ | 0.541** | $y = -52.88x + 328.52$ | 0.4207** |
| *P. fluorescens* | $y = 0.7443x + 52.143$ | 0.0034** | $y = -29.917x + 172.23$ | 0.6704** |

Data used for these analyses is provided in Supplementary data, worksheets 18 and 19.
*** indicates *p* < 0.001, and ** indicates *p* < 0.01.

making the culture medium alkaline. The other three PSBs showed little change under the two conditions. According to the mineralogical properties of the simulant, it is unlikely that the rise of pH values was caused by the decomposition of the simulant minerals, and *B. subtilis* and *B. licheniformis* are believed to secrete alkaline substances when treated with lunar regolith simulant.

To evaluate the responses of different PSBs to salinity and pH factors mathematically and statistically, the concept of growth inhibition ratio (GIR, Supplementary Measurement Methods, section 1) was applied. The higher the GIR is, the more serious impact the simulant has on the inhibition of growth and reproduction. Linear fitting analysis was conducted between the GIR and the soluble salt content at 2, 4, 7, 10, 15, and 21 DAI, as well between the GIR and the pH of the culture medium at 0, 10, 60 HAI, and 5, 8, 15 and 21 DAI. The results are as follows:

As can be seen from the Table 1, the coefficient of determination, or the $R^2$ value, between the GIR of *B. mucilaginosus* and *P. fluorescens* and its corresponding soluble salt content was very low, showing that the growth inhibition of these two PSBs did not respond to the change of soluble salt content. On the contrary, the coefficient of determination between the GIR of *B. megaterium*, *B. subtilis*, and *B. licheniformis* and the corresponding soluble salt content ranged from 0.3566 to 0.6938. All of them were statistically significant (*P* = 0.003, 0.000, 0.004, *n* = 22, 18, 17, respectively), suggesting a certain positive correlation between the two variables, which indicated that the increase of soluble salt content inhibited the growth and reproduction of these three PSBs to some degree.

The inhibitory impact caused by soluble salt concentration may be a combination of high osmotic stress and ionic toxicity. At the beginning of the culture, the medium provided a solution environment with appropriate salt content for microorganisms, which is conducive to maintaining the normal osmotic pressure of cells. With the continuous decomposition of mineral particles by PSBs, insoluble mineral elements were transformed into soluble compounds, which increased the osmotic pressure of the solution and may make PSBs cells unable to maintain intracellular water potential, causing water outflow of cells[44,45], further leading to cell volume contraction. Meanwhile, considering the elemental composition of the lunar regolith simulant, the salts dissolved in the culture medium may contain a large amount of Fe, Al, Mg, K, and Na ions, which could replace the metal ions in the active site of some proteases in the cells[46,47]. This substitution reaction caused adverse changes in the structure of the active sites and thus inhibited their activity. In addition, these ions could induce the production of a large number of reactive oxygen species (ROSs), which further led to severe damage to cell membrane lipids and proteins that were essential to the normal metabolic activities of cells[48,49]. Although microorganisms could resist salt stress through initiative ion transportation[50,51] and synthesis of protective substances (e.g., glycerol[52,53]), growth and reproduction would still be delayed. These factors could explain

the growth inhibition of *B. subtilis* and other PSBs when the soluble salt content increased, which had been also reported by transcriptome data analysis in the study conducted on the real lunar regolith[32].

The linear fitting analysis results of the GIR and the pH of the culture medium showed that, within the range of pH changes produced in the experiment, the coefficients of determination of *B. mucilaginosus*, *B. megaterium*, and *B. subtilis* were very low, which did not exceed 0.1, indicating that the change of pH was not the main factor affecting the growth of these three PSBs. The coefficient of determination of *B. licheniformis* and *P. fluorescence* were 0.4207 and 0.6704 respectively, and both were statistically significant (*p* = 0.001, 0.000, and *n* = 24, 28, respectively), indicating that the decrease of pH inhibited the growth and reproduction of the two PSBs to some degrees. What is more interesting is *B. mucilaginosus*, whose growth and reproduction were not affected by low pH stress or high salt stress. It even grew better in the treatment with the addition of the lunar surface regolith simulant. Based on the ability of *B. mucilaginosus* to decompose silica-containing minerals, we speculated that *B. mucilaginosus* might have a certain tendency to be silicophilic, and the simulant provided silicon element.

In conclusion, under the activities of PSBs, the soluble salt content of the lunar surface regolith simulant increased, which in turn inhibited the growth and reproduction of all PSBs. In addition, the metabolic products of PSBs also changed the pH of the culture medium, which had a negative correlation to the growth and reproduction of *B. licheniformis* and *P. fluorescence* only. Meanwhile, the lunar surface regolith simulant also induced *B. licheniformis* to increase the pH of the culture medium, thus promoting its growth compared with its control after all. The experiment we carried out is still a preliminary study, considering only soluble salt content and the pH of the culture medium were considered. If experimental conditions permit, detailed exploration may further verify the conclusion of this experiment.

**Dynamics of phosphorus content and forms in the lunar regolith simulant treated by PSBs.** The key to the microbial regolith improvement program is to confirm the microbial dissociation and activation of essential mineral elements for plants (e.g., phosphorus) in the lunar surface regolith simulant. In this study, the dynamics of dissolved inorganic phosphorus, dissolved organic phosphorus, suspended microbial biomass phosphorus, attached microbial biomass phosphorus, and adsorbed phosphorus were examined. The classification aimed to study the differences in the spatial and temporal distribution of different forms of phosphorus in more detail. At the beginning of the culture, phosphorus in the medium and inoculation solution was added to the culture system in the form of dissolved phosphorus and microbial biomass phosphorus, and these exogenous phosphorus inputs would be a part of the start-up conditions of the culture process.

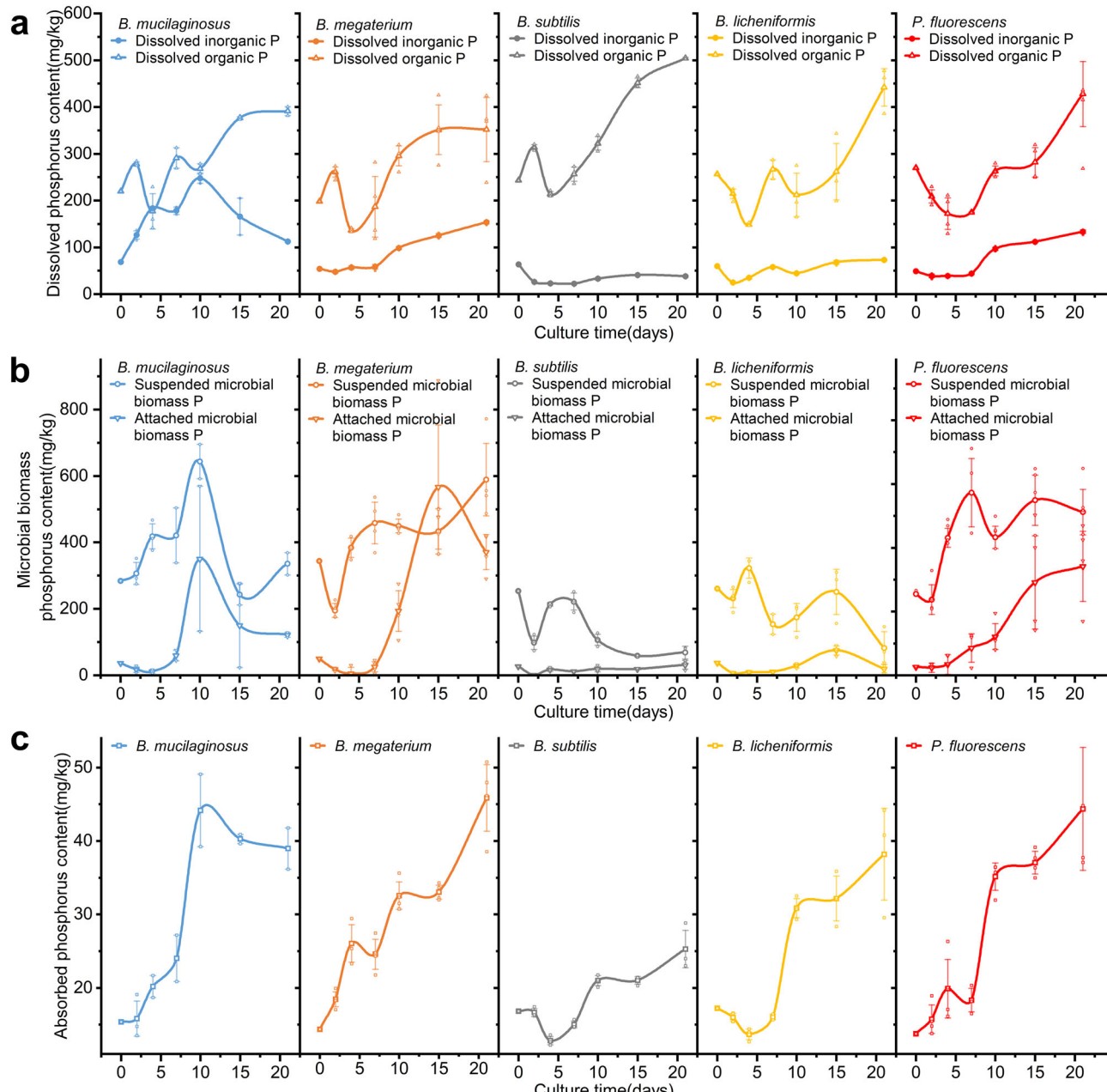

**Fig. 3 Dynamics of the phosphorus content in different forms during the culture. a** The dynamics of dissolved inorganic phosphorus and dissolved organic phosphorus of five PSB treatments. The corresponding control is presented in Supplementary Fig. 1. Each treatment included four flasks as replicates. All error bars represent Standard Deviation. **b** The dynamics of Suspended microbial biomass phosphorus and Attached microbial biomass phosphorus content of five PSB treatments. Each treatment included four flasks as replicates. All error bars represent Standard Deviation. **c** The dynamics of Adsorbed phosphorus content of five PSB treatments. Each treatment included four flasks as replicates. All error bars represent Standard Deviation.

Dissolved phosphorus is the most direct form of phosphorus for plants and microorganisms to absorb, so it is considered as an important indicator for the fertility of lunar regolith simulant. In this study, with the progress of the culture, the overall trend of dissolved phosphorus in regolith simulants with the treatments of five PSBs was increasing. The dissolved phosphorus content at 21 DAI was significantly higher ($p < 0.001$, $n = 21$) than that at the beginning of the culture, ranging from 1.63 times to 2.15 times of the initial value. The difference in dissolved phosphorus content among all five PSBs at the end of culture (21 DAI) was not very obvious compared with that at the middle of culture (4–15 DAI), which may be a sign of sufficient processing. Dissolved phosphorus in the five PSB treatments was mainly organic

(Fig. 3a) at 21 DAI, and dissolved organic phosphorus could reach 2.54 times of dissolved inorganic phosphorus (*B. megaterium* treatment) to 13.20 times of that (*B. subtilis* treatment). This should be the result of the continuous absorption and conversion of inorganic phosphorus into organic phosphorus compounds by PSBs through intracellular biochemical activities. The dissolved organic phosphorus compounds held by PSB cells (e.g., nucleic acid molecules, phospholipid molecules, and adenosine phosphate molecules) can be released back into the culture medium through cell rupture after death.

Microbial biomass phosphorus is an important temporary storage and transfer pool of phosphorus. Phosphorus in minerals can be dissociated and stored in bacteria cells to prevent it from

being fixed again through absorption while the microbes are alive. After they die, the phosphorus in the cells is released and can be absorbed by higher plants or other microbes. In this study, total microbial biomass phosphorus showed a large difference among 5 PSB treatments over the whole culture. For *B. subtilis* and *B. licheniformis*, the microbial biomass phosphorus fluctuated at the initial level and eventually was lower than the initial level at 21 DAI. Considering the obvious increase of dissolved organic phosphorus in the treatment of these two PSBs from 10 to 21 DAI, it can be deduced that a large number of microorganisms died at this stage, and cell rupture caused the transfer of microbial biomass phosphorus to dissolved organic phosphorus. However, the microbial biomass phosphorus of the simulant treated by *B. mucilaginosus*, *B. megaterium*, and *P. fluorescens* maintained higher than the initial level after 2 DAI, suggesting that these three PSBs have a stronger ability to dissociate phosphorus elements in the simulant through microbial metabolic activities, in addition to using phosphorus compounds in the liquid medium. Microbial phosphorus was divided into suspended microbial biomass phosphorus and attached microbial biomass phosphorus according to whether the PSB cells were attached on the simulant particles. Suspended microbial biomass phosphorus was always much higher than attached microbial biomass phosphorus among all five PSB treatments (Fig. 3b), suggesting that the major part of microorganisms in the culture system was in a free state and did not have direct contact with mineral particles. The attached microbial biomass phosphorus of five PSB treatments was very small in 0–4 DAI, while that of *B. mucilaginosus*, *B. megaterium*, and *P. fluorescens* treatments began to increase substantially after 4 DAI. In particular, the attached microbial biomass phosphorus of *B. megaterium* and *P. fluorescens* treatments increased 29.6 times and 6.7 times respectively at 15 DAI compared with that of 4 DAI, and still only accounted for 15.63% to 43.93% of microbial biomass phosphorus.

Adsorbed phosphorus is a dynamic regulating reservoir of phosphorus elements. When the content of dissolved phosphorus in the liquid medium increases, the equilibrium of adsorption-desorption shifts backward and phosphorus tends to be adsorbed. When the phosphorus in the liquid medium decreases with biological absorption and utilization, the adsorbed phosphorus dissolves again, maintaining the relative balance of dissolved phosphorus concentration. In our study, the adsorbed phosphorus of all five PSBs treatments increased significantly ($p < 0.001$, $n = 21$) from the initial value to 21 DAI, ranging from 1.50 times of *B. subtilis* treatment to 3.22 times of *P. fluorescens* treatment (Fig. 3c). During the period of 7–10 DAI, the five PSB treatments all experienced a relatively obvious increase and maintained a relatively stable change in the subsequent culture, which appeared to be similar to the dynamics of soluble salt content, but we are yet insufficient with experimental evidence to find out the relationship between two indexes. After all, the adsorbed phosphorus content was relatively very small in this study, with a maximum of 45.88 mg/kg (21DAI, *P. fluorescens*).

In the control group, which was not treated with PSBs, dissolved phosphorus content remained unchanged, while dissolved inorganic phosphorus content only increased by 4.97 mg/kg after the culture. We believe that there was no spontaneous phosphorus dissociation in the lunar regolith simulants, and the liquid medium could not leach insoluble inorganic phosphorus elements from the simulant, either.

To quantitatively measure the microbial dissociation of phosphorus in the lunar regolith simulant, we defined the concept of total available phosphorus which included all forms of phosphorus, except insoluble phosphorus, representing the sum of initial exogenous phosphorus input (dissolved phosphorus and microbial biomass phosphorus from the medium and the inoculant) and all phosphorus dissociated by PSBs. Statistical analysis showed that the total available phosphorus capacity could be significantly increased by *B. mucilaginosus*, *B. megaterium*, and *P. fluorescens* ($p < 0.001$, $n = 13$), reaching the maximum at 10 DAI, 21 DAI, and 21 DAI, which was 213.57%, 234.31%, and 247.09% of that of the initial level, respectively (Fig. 4A–C). Considering that there was little dissolved phosphorus in the lunar regolith simulant before the culture, the initial total available phosphorus could be considered to be entirely provided by liquid medium and inoculate solution. Therefore, the extremely significant increase in the total available phosphorus capacity during the 21-day culture process could be completely regarded as the result of the dissociation of insoluble inorganic phosphorus elements in the simulant by the three PSBs. The amounts of insoluble inorganic phosphorus dissociated by *B. mucilaginosus*, *B. megaterium*, and *P. fluorescens* at 21 DAI were 374.75 mg/kg, 887.46 mg/kg, and 904.03 mg/kg, respectively (Fig. 5). In contrast, *B. subtilis* and *B. licheniformis* did not significantly ($p = 0.141$, $n = 8$) increase the capacity of total available phosphorus during the culture (Fig. 4D, E), and even caused a relative decrease of that during 0-10 DAI. The total available phosphorus of the two PSB treatments at the end of the culture was only 1.11 times and 1.16 times of the initial level, so it could be considered that the ability of the two PSBs to dissociate insoluble inorganic phosphorus in the simulant was very weak.

From the perspective of the forms of phosphorus, the dissociation of phosphorus from the lunar surface regolith simulant of *B. mucilaginosus*, *B. megaterium*, and *P. fluorescens* treatments during the 21-day culture was analyzed (Fig. 4F). In *B. mucilaginosus* treatment, dissolved organic phosphorus was the most important factor for the increase of total available phosphorus capacity, accounting for 45.70% of the total increase. This indicated that *B. mucilaginosus* may have a strong ability to excrete organic matter that contains phosphorus compounds, which is worth further investigation. The main contributing factor for *B. megaterium* and *P. fluorescens* treatments was microbial biomass phosphorus, accounting for 67.70% and 69.80% of the total increase, respectively. Among the three PSB treatments, the contribution of adsorbed phosphorus to the total increase was very small, ranging from 6.30% of *B. mucilaginosus* treatment to 3.39% of *P. fluorescens* treatment, indicating that adsorbed phosphorus was not an important factor.

**The mechanisms of the microbial dissociation of phosphorus in the lunar regolith simulant**. In this study, the mechanisms of the dissociation of insoluble inorganic phosphorus were explored from the perspective of the pH of the culture medium and bacterial attachment on regolith simulant particles. Considering that the pH of the culture medium of five PSB treatments remained basically stable during 7–21 DAI, and the dissociation of inorganic phosphorus is a continuous process, the average pH of 7 DAI, 15 DAI, and 21 DAI were chosen as the independent variable, the amount of insoluble inorganic phosphorus dissociated by PSBs (DIIP, mathematically equals to the difference between the total available phosphorus and the initial phosphorus input) at 21 DAI as the dependent variable, and carried out a linear fitting analysis (Table 2). the coefficient of determination was 0.8362 and reached an extreme statistical significance ($p < 0.001$, $n = 16$). We believed that the decrease of medium pH is an important factor for the PSBs to dissociate insoluble inorganic phosphorus in the simulant.

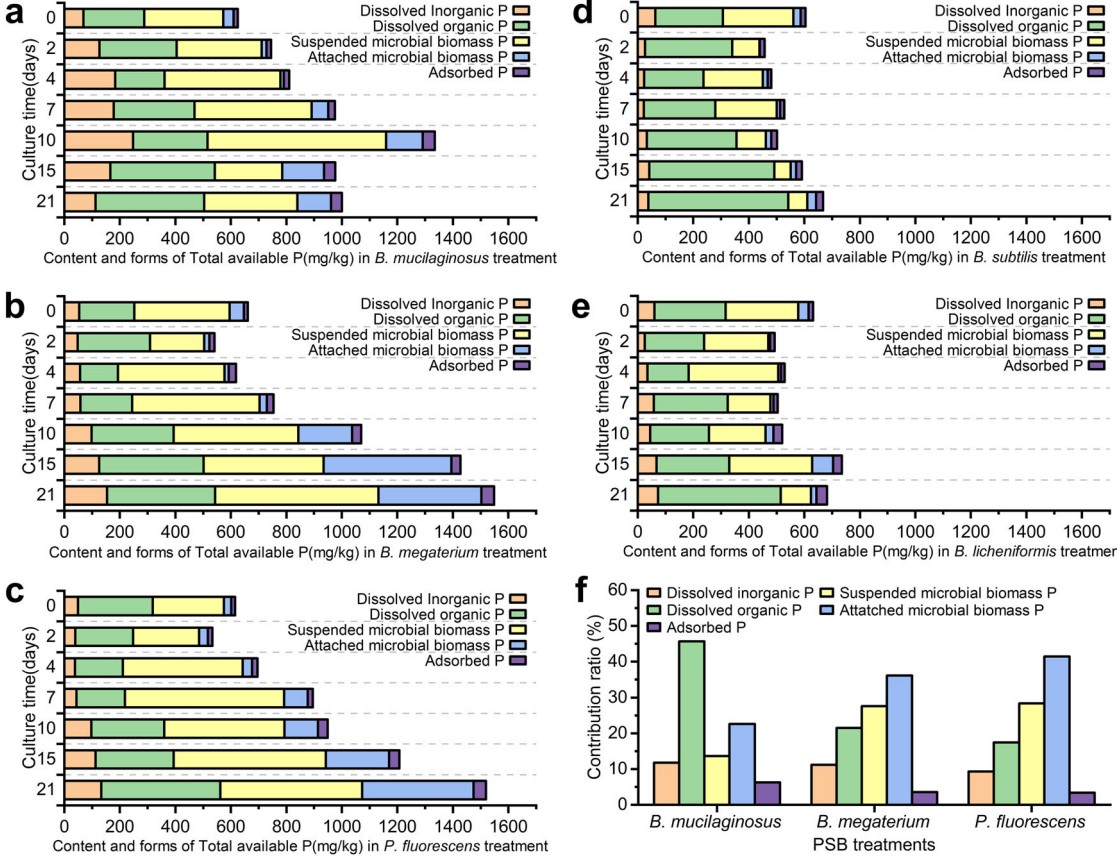

**Fig. 4 Dynamics of the content and forms of total available phosphorus in five PSB treatments. a–e** Represent *B. mucilaginosus, B. megaterium, P. fluorescens, B. subtilis,* and *B. licheniformis* treatments respectively. **f** The contribution ratio of different forms of phosphorus during the change of total available phosphorus capacity. All figures in Fig. 4 are plotted according to the average value of different forms of phosphorus, therefore no error bar is applicable.

The relationship between low pH and insoluble inorganic phosphorus dissociation has been confirmed by a large number of studies so far. Many studies believe that the acidity enhancement of the culture medium is an important feature of microbial phosphorus solubilization[54–56]. Lin et al. suggested that the reduction of pH is an important condition for microbes to dissociate inorganic phosphorus[57], although not necessary. In the culture system, protons come from two sources: one is from the metabolic activities of cells, such as $CO_2$ produced by respiration would form carbonic acid when dissolved in water, or the by-product of absorption and utilization of $NH_4^+$[58,59]; The other is organic acids produced and secreted by bacteria to the extracellular environment[60–63] (e.g., malic acid, oxalic acid, citric acid, tartaric acid, etc.). Phosphoric acid ions are released when protons exchange with mineral crystals. Meanwhile, according to the research results based on the Earth soil, protons can promote the decomposition of $CaCO_3$ in calcitic soil, while Fe, Al oxides, and hydrating oxides can be decomposed in red soil, thus greatly reducing the fixation of phosphorus in soil[64–66]. We believe a similar process occurred on the lunar surface regolith simulant. In addition, organic acid ions also facilitate the dissolution of phosphate minerals through chelating with the metal ions in $Ca_3(PO_4)_2$, $FePO_4$, $AlPO_4$, and other insoluble phosphorus-containing compounds, resulting in the release of phosphorus acid ions. This mechanism has been confirmed by experimental evidence[67–70]. The type and quantity of organic acids are believed to be critical to the chelation ability with metal ions[71,72], and this may be the reason that different PSBs differ in their ability to dissociate insoluble inorganic phosphate in regolith simulants.

In addition, linear fitting analysis was performed between the amount of attached microbial biomass phosphorus and DIIP at 0, 2, 4, 7, 10, 15, and 21 DAI, with the coefficient of determination of five PSB treatments ranging from 0.5262 to 0.8597, all reaching extreme significance ($p < 0.001$, $n = 14$, 24,19, 19, 23, respectively). These results indicated that the attachment of PSBs to the regolith simulant particles was also an important factor in the dissociation of insoluble inorganic phosphorus. Considering that *B. mucilaginosus, B. megaterium,* and *P. fluorescens* that showed strong phosphorus dissociation ability in this study also had a strong ability to secrete extracellular sticky substances such as extracellular polysaccharides[73] to form biofilms, we speculated that these three PSBs may be able to bind themselves with mineral particles through biological adsorption and developing biofilms to tightly bind with mineral particles, forming a stable bacteria-mineral complex[74] and a relatively stable extracellular microenvironment was formed in the biofilm, where bacteria can better decompose and dissociate mineral particles through the above-mentioned mechanisms, including protons and organic acids.

In conclusion, we confirmed that *B. mucilaginosus, B. megaterium,* and *P. fluorescens* have a strong ability to dissociate insoluble inorganic phosphorus in the lunar regolith simulant due to their ability to secrete acidic substances and their characteristics to attach with simulant particles. On the contrary, *B. subtilis* and *B. licheniformis* had poor performance in these two aspects, so they had little dissociation activity on insoluble inorganic phosphorus in the regolith simulant. It can be inferred that the lunar regolith simulants reduced their ability to secrete acidic

substances, thus making them unable to dissociate insoluble inorganic phosphorus.

**Plant growth in the lunar surface regolith simulant treated with PSBs.** Based on the results above, *B. mucilaginosus*, *B. megaterium*, and *P. fluorescens* were selected for subsequent experiments on plant growth. The cultivation experiment was conducted on *Nicotiana benthamiana*. Most seeds had sprouted two cotyledons at 6 days after sowing (DAS). At 21 DAS, there were still 20–40 available plants for each treatment, which could be used as samples for measurement and analysis.

During the cultivation, seedlings in two control groups showed obvious growth inhibition and slower growth, compared with *Nicotiana benthamiana* planted in horticultural soil. At 24 DAS, the average fresh weight of whole plants (including root) in blank and sterilized control groups was only 23.48% and 29.83% of that

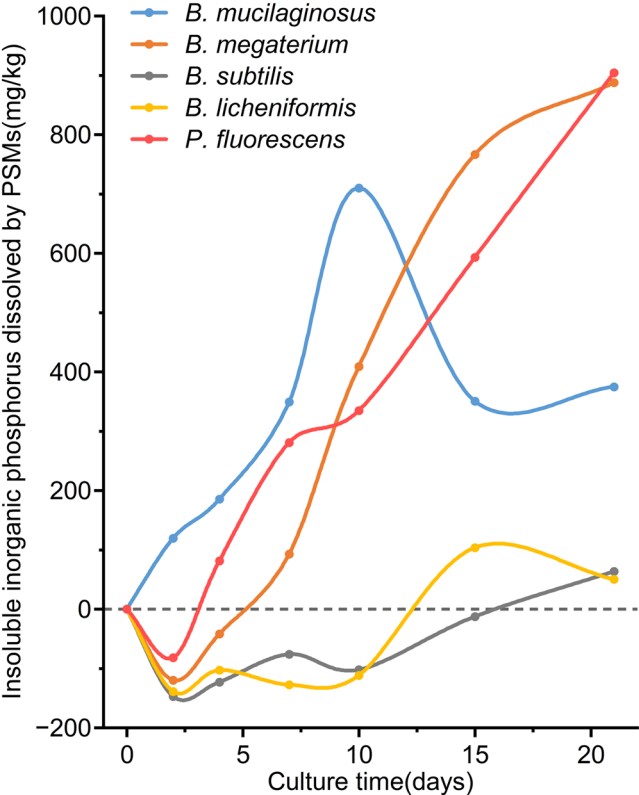

**Fig. 5 Dynamics of dissolved insoluble inorganic phosphorus (DIIP) content of five PSB treatments.** This figure is plotted according to the average value of different forms of phosphorus, therefore no error bar is applicable.

in the horticulture soil group, respectively, and all reaching extreme statistical significance ($p < 0.0001$, $n = 56$, 33). The regolith simulant in both control groups, which lacked microbial treatment, is likely to have some negative effect on the growth and development of higher plants, or it may just be delayed by the poor fertility of the simulant.

Other results showed that this negative effect can be eliminated by PSBs treatment. The rosette leaves diameter at 24 DAS, length of hypocotyl and radicle of the seedlings at 6 DAS, fresh weight per plant, and chlorophyll content of *Nicotiana benthamiana* plants at 24 DAS (Fig. 6b, c) were measured, as important indicating indexes of plant growth. It is worthwhile mentioning ahead that we did not observe a significant difference in rosette leaves diameter and chlorophyll content in the blank and sterilized control groups ($p = 0.0668$, 0.3454, $n = 56$), while the other two indexes had a small difference in their average values with significance ($p = 0.0036$, 0.0430, $n = 56$). We may conclude that the nutrients in dead microbes and broth medium of the sterilized control group had little impact on plant growth in this experiment, so it was needless to be considered in the subsequent analysis.

Then, the differences in indexes in different treatments were examined. Among all four indexes that we had measured, we would firstly focus on the content of chlorophyll in *Nicotiana benthamiana* plant leaves, as chlorophyll plays the most central role in the light reaction of photosynthesis, and is widely considered an important indicator of plant growth. It is obvious in Fig. 6c that the chlorophyll content in the "Pre-cultured for 18 days" treatment is 104.08% higher than that in the sterilized control, reaching an extreme statistical significance ($p < 0.0001$, $n = 53$). Meanwhile, the difference between the "Pre-cultured for 18 days" treatment with the horticultural soil group was not significant ($p = 0.1580$, $n = 44$). This experimental evidence shows that the growth of plants in the lunar regolith simulant pre-cultured by PSBs for 18 days had been greatly improved, reaching the level that was as good as horticultural soil on Earth. The fresh weight of the plant in the "Pre-cultured for 18 days" treatment, which was also significantly higher than that of the sterilized control ($p < 0.0001$, $n = 53$), also reflected the result of the active and efficient photosynthesis.

We observed the overall trend that the growth of plants was improved as the time of the pre-culture period extended, as Figs. 6b, c, 7, and the Table 3 represent.

The properties of the lunar regolith simulant samples at different times during the culture process were examined to confirm that the PSBs promoted plant growth. Similar to the flask shaking experiment, the pH and $OD_{600}$ of the regolith extract, the available phosphorus content, and the soluble salt content of the simulant (Fig. 6d–g) was measured. The most noticeable changes occurred within six days of starting the culture when the OD600 reached its peak and the regolith pH and available phosphorus

**Table 2 Results of linear fitting analysis between the amount of DIIP and the pH of the culture medium and the attached microbial biomass phosphorus.**

| PSBs | Linear fitting equation between DIIP and the pH | $R^2$ | Linear fitting equation between DIIP and the attached microbial biomass phosphorus | $R^2$ |
|---|---|---|---|---|
| *B. mucilaginosus* | $y = -220.27x + 1838.1$ | 0.8362*** | $y = 1.8696x + 139.79$ | 0.8567*** |
| *B. megaterium* | | | $y = 1.6485x + 12.589$ | 0.8512*** |
| *B. subtilis* | | | $y = 4.9942x - 148.97$ | 0.5721*** |
| *B. licheniformis* | | | $y = 2.4674x - 131.53$ | 0.5262*** |
| *P. fluorescens* | | | $y = 2.1034x + 15.736$ | 0.8597*** |

Data used in these analyses are provided in Supplementary data, worksheets 20 and 21.
*** indicates $p < 0.001$.

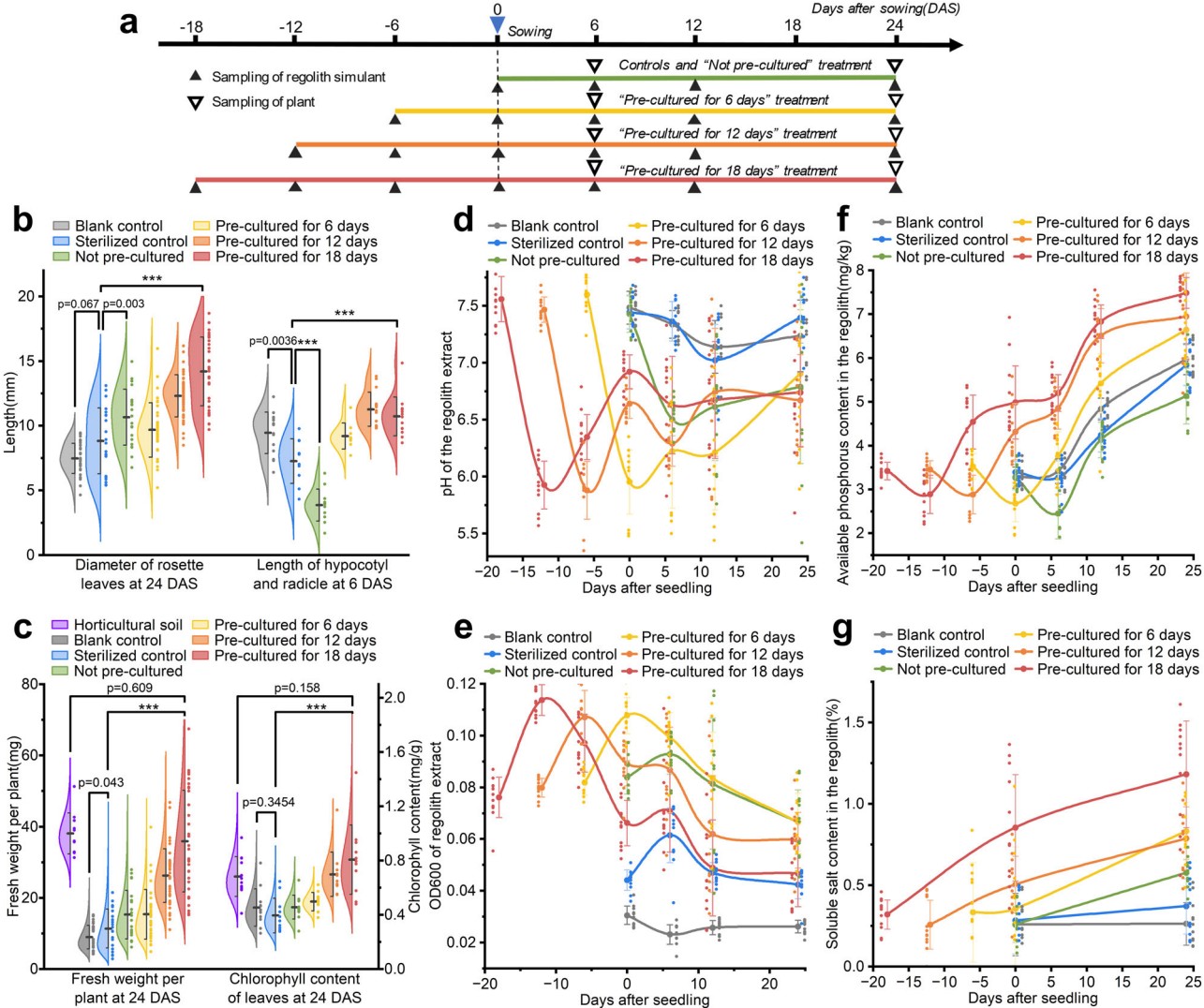

**Fig. 6 The Cultivation experiment of *Nicotiana benthamiana*. a** The workflow of the whole cultivation experiment, showing all sampling procedures on the time axis of days before/after sowing. Negative numbers imply the number of days before sowing. **b** The measurement results of the Diameter of rosette leaves at 24 DAS, and the length of the seedlings at 6 DAS. All error bars in Fig. 6 represent Standard Deviation. Significance analyses are conducted using one-side one-way ANOVA tests. *** indicates $p < 0.0001$. **c** The content of fresh weight of whole plant (root included) and the chlorophyll content of leaves at 24 DAS in different treatments. **d–g** The dynamics of pH and the $OD_{600}$ of the regolith extract (regolith-water mass ratio was 1:2), the available phosphorus content, and the soluble salt content in different treatments during the cultivation. The data points (small points) of different treatment groups are offset horizontally, with the average value point (large points) as the center, to avoid overlap between data points.

content dropped to their lowest. Based on the results of the shaking flask experiment, it is easy to understand that the PSBs firstly used the nutrients in the medium to grow, consuming the phosphorus in it. The low phosphorus content then promoted PSBs to synthesize organic acids by using carbon sources in the medium to reduce regolith pH, thus activating insoluble inorganic phosphorus in lunar regolith simulant. After 6 days, the $OD_{600}$ dropped and the regolith pH began to rise in all four treatments, we suspect that this may be due to the consumption of organic acids by PSBs as a carbon source for growth. Meanwhile, we observed a steady increase of available phosphorus content before 6 DAS, with a regolith pH lower than 7 in all four PSBs treatments. The phosphorus content was largely determined by the addition of Murashige & Skoog medium after 6 DAS, which added 1.27 mg/kg phosphorus to the simulant every 6 days. It could be concluded that microbial reactions in the cultivation experiment followed the same pattern that we concluded before in the shaking flask experiment, in which the

acidic substances were the key factor in dissolving inorganic phosphorus from the simulant.

The result of the "Not pre-cultured" treatment was more interesting. In this treatment, we observed an extremely significant decrease in the length of seedlings at 6 DAS, compared with sterilized control ($p < 0.0001$, $n = 22$). The average was only 53.28% of that in sterilized control. However, plants in this treatment did not show such extreme differences with the control on other indexes, and we are yet unable to explain this decrease with our data on the properties of soil samples. We may need more careful examinations to explain this phenomenon.

The content of soluble salt in the regolith was also examined, but we did not find any related growth inhibition in plants or PSBs. The *Nicotiana benthamiana* plants grew better as the time of the pre-culture period prolonged, while the content of the soluble salt in the regolith also increased. We predicted that the positive impact brought in by the pre-culture period, maybe extra nutrients, or other microbial derived plant hormones,

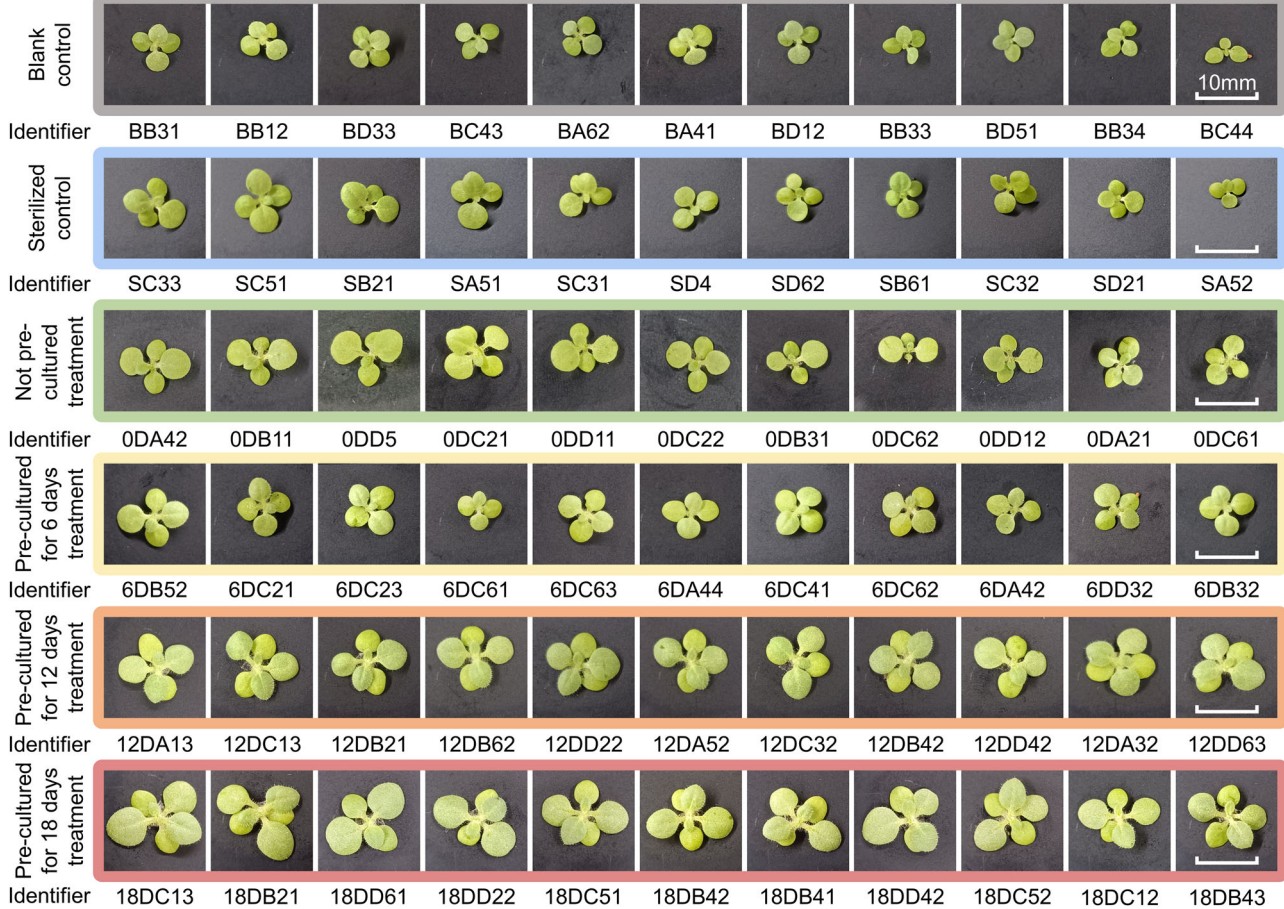

**Fig. 7 The photography of partial plants in two control groups and four treatments.** We selected 11 images for each group of plants, including plants at the median leaf diameter of each treatment in the middle of the row, as well as five slightly larger and five slightly smaller plants on the left and the right. Each row of images shares one ruler, which is given in the image on the far right. The identifier of each image is given at the bottom of each image. Other images are available upon request.

**Table 3 Comparison of four growth indexes between Sterilized control(setting the value as 100%) and other treatments.**

| Treatments | Diameter of rosette leaves at 24 DAS | Length of seedling at 6 DAS | Fresh weight per plant at 24 DAS | Chlorophyll content at 24 DAS |
|---|---|---|---|---|
| Sterilized control | 100% | 100% | 100% | 100% |
| Not pre-cultured | 120.86% | 53.28% | 134.52% | 115.52% |
| Pre-cultured for 6 days | 109.62% | 126.62% | 135.39% | 125.80% |
| Pre-cultured for 12 days | 139.49% | 155.14% | 230.83% | 176.80% |
| Pre-cultured for 18 days | 160.86% | 147.53% | 315.92% | 204.08% |

counteracted the negative impact that soluble salt content has on plant growth. We look forward to explaining this in the next phase of research.

The cultivation experiment of *Nicotiana benthamiana* further confirmed the ability of PSBs to dissociate insoluble phosphorus in lunar regolith simulant, making it a cultivation substrate for higher plants and supporting their growth. The results of the cultivation experiment showed that the growth of *Nicotiana benthamiana* could be most improved by treating the regolith simulant with PSBs 18 days before sowing. Meanwhile, the experiment revealed the negative impacts caused by short treatment of PSBs, which may be explained in further studies.

**Application prospect of PSBs for lunar exploration and settlement.** Looking beyond the perspective of microbiology and relooking our study from the perspective of lunar exploration and

settlement, it is not difficult to understand the strong potential of PSBs to achieve sustainable space agriculture in the BLSS and to improve the recycling capacity of manned closed ecosystems on the moon in the further future, without bringing too much burden in the transportation. A lunar base, as a closed, self-sustaining artificial ecosystem, would require a constant exchange of matter to keep it from collapsing and to support the survival and daily activities of human crews. In the simplest model, the Earth provides all the materials needed for the human crew to survive: water, food, and air, just as we did in low-Earth orbit when space exploration just began. As we have argued in the introduction, the costs of these operations would be acceptable (though still very high) for low-Earth orbit, but unaffordable for long-term exploration plans towards more distant targets (e.g., the moon, Mars, etc.) The introduction of photoautotrophs into an artificial ecosystem is the first great step to try to resolve this problem. The

great value of algae[15,75,76], or higher plants[77], has been repeatedly demonstrated by studies in various countries for their ability to purify the atmosphere, and water, and provide food. It has become a consensus that a well-designed and carefully balanced BLSS system supported by photoautotrophs is sufficient to meet the oxygen requirements of human crews over long-term scales, as well as to fix carbon and partially provide food through photosynthesis. By doing so, a material circulation system of two components can be established within the closed artificial ecosystem, reducing the need for material exchange with the Earth.

But, looking at it another way, we are simply converting the need to transport food, water, and air into the need to build an extraterrestrial system of plant growth, including cultivation substrate, nutrients, water, and equipment. Given the huge requirements of existing designs on transportation capacity and cost[18–25], unless such a system can be self-sustaining over very long periods, the net benefit remains very poor in the short term. Introducing bacteria—micro decomposers of nature—will be the key to solving this problem. Bacteria can transform the most abundant resource of the moon—surface regolith or rocks, which has undergone a lengthy space weathering process, into an active, bio-friendly, and moderately fertile cultivation substrate, eliminating the need for complex extraterrestrial hydroponic systems, and this possibility has been demonstrated by our experiments.

The even more exciting thing is that the applications of PSBs also leave room for organic integration in another area of microbial-mediated technology: soil-like substrate. The soil-like substrate is a technology that uses organic waste including food residues, human excrement, and plant residues to produce cultivation substrate which is similar to earth horticultural soil and requires the participation of microbial fermentation and soil animals (e.g., earthworms[78]). Basically, it provides a method to recycle organic wastes in the extraterrestrial closed ecosystem. From the perspective of means of making cultivation substrates, soil-like substrates still have a series of shortcomings. For example, soil-like substrates often experience a large degree of dry matter mass loss during the producing period[78,79], making them unable to cope with long-term plant cultivation needs. However, in conjunction with the process by which PSBs treat the lunar surface regolith, we can imagine like this: In a future lunar base, all organic waste is composted in closed containers. The compost products and the inoculation solution of PSBs will be applied to untreated lunar surface regolith, which has been pre-sifted to proper particle size. Here, PSBs use organic matter in compost products as a nutrient source and decompose mineral crystals in the regolith to further enhance the fertility level of the regolith. Meanwhile, compost products, as an organic fertilizer, increased the carbon content of lunar regolith, improving its physical and chemical properties and avoiding its negative influence of low output. By doing so, the demand for exogenous nutrient solutions in culturing PSBs and cultivating higher plants has been eliminated. It is reasonable to speculate that, in combination with the PSBs process and the soil-like substrate fermentation, the lunar regolith will become a cultivation substrate that is very similar to horticultural soils on the Earth and suitable for the growth of most agricultural higher plants. This would greatly facilitate practices of space agricultural systems in lunar bases without additional transportation requirements.

However, considering the concerns on the potential biosafety of microorganisms, it is necessary to conduct a thorough pathogenic examination of any microorganisms introduced into an extraterrestrial closed artificial ecosystem to ensure that their biosafety risks to other organisms are fully controlled and do not pose a threat to human crews. Synthetic biology may provide a useful opportunity to create artificial life with efficient

phosphorous decomposition abilities by molecular biological methods based on understanding the mechanism by which PSBs and other microbial fertilizers improve the fertility of lunar regolith, and ensure its existence being strictly restricted to extraterrestrial closed artificial ecosystems.

In conclusion, our study shows that *B. mucilaginosus*, *B. megaterium*, and *P. fluorescens*, can tolerate lunar regolith simulant conditions and effectively dissociate insoluble inorganic phosphorus in the simulant, improving the fertility of the simulant making it a good cultivation substrate for higher plants. We have therefore proved that it is feasible for PSBs to improve lunar regolith, and PSBs have great application value and prospects for future space exploration.

## Methods

**Lunar surface regolith simulant materials**. The lunar surface regolith simulant used in this study was modeled from the CAS-1 lunar soil simulant developed by Zheng et al. [80], and the reference object was the lunar low-titanium basalt regolith. According to the study of Zheng et al., the scoria of Jinlongding Volcano in the Longwan volcanic Group at the western foot of Changbai Mountain (located near Huinan County, Jilin Province, PRC) has a very similar chemical composition and physical structure to the lunar low-titanium basalt regolith and contain volcanic glass to a degree of around 40%, which has a good simulation effect on the lunar regolith. The initial material of the lunar regolith simulant used in this study was collected from about 3 km east of the Jinlongding volcanic cone, and the thickness of the scoria layer at the sampling site was about 2 m and the sampling depth was 1.5 m. The surface of the sampling site was protected by thick vegetation, thus preventing the scoria from the direct wash of rain. The collected scoria is black porous trachyte basalt, about 1–2 cm in diameter. Scoria was then sufficiently broken up and sifted to produce simulants with a range of particle sizes. In the pre-study, it was found that: <0.3 mm simulation was easy to cause turbidity in the shaking flask culture, which added difficulty to analysis; In addition, the study of Paul et al. [33]. showed that when the soil with a large proportion of fine particles was used in the cultivation of Arabidopsis, the soil showed a poor porosity and water permeability, which could easily lead to heavy soil viscosity and plant root dysplasia. Therefore, the particle size of the simulant used in this study was 0.3–0.5 mm.

The samples were then digested and the mass ratio of Fe, Ca, Mg, Na, K, P, and Mn was determined using an atomic flame spectrophotometer. Three replicates of our samples were used for measurement. This part of the determination was completed by the Experimental Center of the College of Agriculture, China Agricultural University.

**Bacteria materials**. The PSBs used in this study were selected based on their ability to activate insoluble inorganic phosphorus in Earth soils. According to the current research results[81–88], five PSBs were used in this study: B. megaterium (AS1.217), B. subtilis (CMCC 63501), P. fluorescens (ATCC 13525), B. licheniformis (ATCC 11946), and B. mucilaginosus (AS1.232), which are commonly used as microbial fertilizers. All PSBs were provided by the Shanghai Bioresource collection center. The PSBs were first inoculated on agar medium plates for activation. After an incubation of 24 h at 30 °C, single colonies with typical morphological characteristics were picked and inoculated into 30 ml of liquid medium for expanded culture. The strains were cultured overnight in a shaking flask at 180 rpm and 30 °C. Subsequently, the $OD_{600}$ value of the culture was adjusted to 0.6–0.8 with sterile water (0.15 for *B. mucilaginosus*, because its cells were more transparent), and the CFU of the solution was about $1 \times 10^8$,

which could be used as the inoculation solution for subsequent experiments. It should be noted that the culture medium for *B. mucilaginosus* was silicate bacteria medium (glucose 5 g/L, $MgSO_4$ 0.5 g /L, $CaSO_4 \cdot 2H_2O$ 0.1 g/L, $Na_2HPO_4$ 2 g/L, $FeCl_3$ 0.005 g/L, pH = 7.0), as the provider of *B. mucilaginosus* suggest that it would have maximum growth in silicate bacteria medium. The medium for other PSBs was the LB medium. Solid medium was made by adding a 2% mass ratio of agar to the liquid medium. All mediums were sterilized at 115 °C for 15 min and tested for sterility before use.

**Plant materials**. The seeds used in the cultivation experiments were provided by Professor Ronghui Pan, Zhejiang University, PRC. The seeds were collected in June 2022.

**Standard test for the ability of microbes to dissolve insoluble inorganic phosphorus**. To ensure the validity of the experiment, the ability of the five strains to dissolve insoluble inorganic phosphorus was tested using standard methods before inoculating them with lunar regolith simulant. The inoculation solution of five strains was inoculated to 30 ml liquid medium (glucose 10 g/L, $(NH_4)_2SO_4$ 0.5 g/L, $Ca_3(PO_4)_2$ 5 g/L, NaCl 0.3 g/L, $FeSO_4$ 0.03 g/L, KCl 0.3 g/L, $MgSO_4$ 0.3 g/L, $MnSO_4$ 0.03 g/L, pH = 7.0), using sterile water as control, shaking flask culture at 30 °C, 180 rpm for 7 days. The culture medium was sampled right after the inoculation and after the culture process. The soluble phosphorus content was measured by the Mo-Sb colorimetric method.

**Shaking flask experiment of PSBs**. A suitable microbial culture system was established after a series of preliminary experiments. The culture medium included two parts: the solid phase and the liquid phase. The solid phase consisted of 40 g of the lunar surface regolith simulant (0.3–0.5 mm) which was pre-dried to constant weight at 70 °C in a 150 ml flask. The liquid phase consisted of 80 ml glucose broth medium (peptone 10 g/L, yeast extract 3 g /L, NaCl 5 g/L, glucose 5 g/L, pH = 7.0). The preliminary test showed that the growth of *B. mucilaginosus* was poor when the LB medium was used (only the change of growth curve was considered). However, all five PSBs had better growth performance when using a glucose broth medium. Therefore, to ensure the uniformity of test conditions, 5 PSBs were cultured with glucose broth medium. The preliminary experiment also found that mixing the solid phase with the liquid phase before using steam sterilization would cause obvious Browning of the medium, and the sterilization effect was poor (the reason was unknown). Therefore, we first sterilized the solid phase at 180 °C dry heat for 3 h, then steam-sterilized the liquid phase at 115 °C for 15 min, and then mixed 80 ml of liquid phase with 40 g of solid phase within an ultra-clean workbench, and sealed a sealing film of air-permeability. Before inoculation, the mixed culture medium was cultured at 30 °C and 180 rpm for 24 h to check whether it was contaminated. The culture medium without contamination will be inoculated with 0.1 ml inoculation solution, and continue shaking culture under the above conditions. In the control groups of the growth experiment, the simulant was not added to the liquid phase. In the control group of the phosphorus measurement, the medium mixed with simulant was not inoculated with PSBs. Each treatment included four flasks as replicates.

We have conducted a series of pre-experiments with different amount of regolith added to the culture, including half of the weight of the culture to ten times of the weight of the culture. We observed that: when the amount of regolith was more than the culture, shaking the culture to maximum the microbial explosure to the oxygen was impossible. Manual stirring of the culture may introduce subjective errors, like the time or the degree of the stirring procedure.

An extreme is adding culture to the regolith to a water ratio of ~20%. The moisture in the regolith allows the regolith to form a relatively loose structure, which is conducive to microbial respiration. However, fewer culture medium prevented microorganisms from further growing, and the regolith was treated less adequately, as we have observed in the preparation of cultivation substrate. The other extreme is adding a small amount of regolith to the culture, which was described in reference 39 & 41. We believed that such treatments are less practicle to be performed on the moon, as water is predicted to be very rare and valuble on the moon. Thus, we finally chose the ratio of 1 : 2, which enables the microbe to be shaking-flask cultured and also a more adequate treatment of regolith by PSBs.

We did not perform toxicity test, as we read from reference 39 & 41 that untreated lunar regolith simulant are "inert" and less likely to affect biological activities, neither positive nor negative. In fact, we observed from our experiment that the regolith simulant treated with different PSBs may show different impact on microbial growth with different mechanisms (salt content, pH), at a addition ratio of 1 : 2. We may study this toxicity more systematically in the future.

**Sampling method of the shaking flask experiment**. We developed and optimized a complex but meticulous sampling method for a solid-liquid mixture shaking flask culture system. Corresponding to the culture medium, the sample is divided into two kinds including solid sample and liquid sample. Considering that both solid and liquid phases have experienced sufficient shaking during the culture process, we believe that the solid and liquid phases have been fully mixed, and their chemical compositions and properties are uniform in space. Before sampling, the culture flask stopped shaking and was let still for about 1 minute to make a full precipitation of the solid phase, separated from the liquid phase completely. Then transfer the flask to the ultra-clean workbench for sampling.

Liquid sampling. Transfer 500 µl of the culture to a marked and sterilized centrifuge tube with a pipette; Dilute the sample if it is necessary for a subsequent determination, and modify the effect of dilution on concentration during analysis.

Solid sampling. Use a sterilized micro U-shaped medicine spoon to scoop 2 g of the precipitated lunar regolith simulant at the bottom of the flask and put it into a marked and sterilized centrifuge tube.

Sampling schedule. Solid and liquid samples were sampled at 0, 2, 4, 7, 10, 15, and 21 DAI. Additional liquid samples were sampled at 0, 10, 17, 26, 32, 48, 60, 72, 84, and 96 HAI, and 5, 6, 7, 8, 9, 10, 15, and 21 DAI for the determination of growth curves.

Sample storage. Both liquid and solid samples were stored at 4 °C and analyzed within 12 h after sampling.

Laboratory measurements of samples. The detailed methods of laboratory measurements, including the measurement of dissolved inorganic phosphorus, dissolved organic phosphorus, microbial biomass phosphorus, suspended microbial biomass phosphorus, attached microbial biomass phosphorus, and adsorbed phosphorus of five PSB treatments, are provided in Supplementary Measurement Methods.

**Preparation of cultivation substrate**. In the cultivation experiment, the substrates of experimental treatments were treated with PSBs before sowing. According to the shaking flask culture experiment results, *B. mucilaginosus*, *B. megaterium*, and *P. fluorescens* were selected as treatment strains.

We start the preparation from the inoculation solution. The three strains mentioned above were cultured overnight in a shaking flask at 180 rpm and 30 °C (described in Bacteria Materials). Subsequently, the CFU of the solution was adjusted with glucose broth medium (described in the Shaking flask experiment of PSBs) to about $1 \times 10^8$, and then the culture of three PSBs were mixed at a volume ratio of 1:1:1. The mixture would be used as the inoculation solution for subsequent experiments.

The experimental treatment was divided into four treatments (Not pre-cultured, and pre-cultured for 6, 12, and 18 days) with 12 replicates each. Twelve 50 ml bottles were filled with 20 g of sterilized lunar surface regolith simulant, 0.1 ml mixed inoculation solution was added at a 0.5% mass ratio, and 1 ml glucose broth medium was added at a 5% mass ratio. Then add sterilized $ddH_2O$ to the water ratio of 20%. The medium was mixed thoroughly and evenly, sealed with a sealing film of air-permeability, and incubated at 30 °C away from light. The simulant was remixed every 2-3 days so that the phosphorus solubilizing bacteria were fully exposed to oxygen for normal respiration. Three bottles were taken out at 0 DAI, 6 DAI, 12 DAI, and 18 DAI, respectively. They were then immediately filled to a 24-well plate for further cultivation. Twelve of the 24 wells are filled alternately. Each well was filled with 4.0 g of simulant (wet weight).

The control group included blank control and sterilized control with 12 replicates each. In the blank control, only sterilized water was added. In sterile control, 1 ml glucose broth medium and 0.1 ml mixed steam-sterilized inoculation solution were added to the lunar surface regolith simulant for every 20 g. Two control groups were not pre-cultured.

**Nicotiana benthamiana cultivation experiment.** A 24-well culture plate was used as the cultivation vessel with a diameter of 17.5 mm and a depth of 17.5 mm for each well. The substrate is saturated by wetting it with sterilized water until a very thin layer of water can be maintained on the surface. *Nicotiana benthamiana* seeds were sterilized and 6-7 seeds were sown in each well. Cover the culture plate with the cover, and promote germination in the artificial climate room. The growth conditions were as follows: temperature 24 °C, relative humidity 70%, light intensity 130 μmol/(m²·s), photoperiod 16/8 h. *Nicotiana benthamiana* seedlings were artificially thinned after germination at 6 DAS, remaining 2–4 plants for each well. The thinning was random, to avoid the influence of subjective criteria on experimental results. From 6 to 24 DAS, 30 μl Murashige & Skoog medium was added to each well every 6 days. The plants were watered daily until the substrate was basically saturated.

**The sampling method of the cultivation experiment.** The samples of the cultivation experiment were divided into two parts: regolith samples and plant samples. Regolith samples were taken with a disposable single-use plastic sampling tube. When sampling, the tube would be inserted directly into the bottom of the soil, and ~0.3 g regolith substrate would remain in the tube when the tube pulls out. The samples would be marked, stored at 4 °C, and analyzed within 2 days.

Plant samples are taken at 6 and 24 DAS. When sampling, the root of the plant would be carefully removed from the soil, and washed with $ddH_2O$.

The detailed methods of laboratory measurements are provided in Supplementary Materials, Measurement Methods.

**Software.** In this study, all the original data determined by Multiskan GO (Thermo Fisher Scientific) were outputted and stored by SkanIt software. SPSS and Excel are used for data sorting and simple statistical description. The correlation analysis, linear regression analysis, and significance analysis of data are completed in SPSS. OriginLab is used in data visualization.

**Statistics and reproducibility.** The numbers of data used in the analyses (n) are shown together with the *p*-value. In this study, we used four replicates for the culture of PSBs and twelve for the cultivation of *Nicotiana benthamiana*. However, due to pollution or other factors, some replicates were stopped from culture or cultivation. The decision was made on the status of each replicate, including the color of the culture, spontaneous precipitation of cells, growth of plants, or data from the measurement. The full description of replicate numbers (and corresponding data numbers used for analysis) at each time of sampling is provided in Supplementary Table 1. In our study, the average value and the standard deviation of the data were calculated. When comparing differences between data groups, one-way ANOVA tests were performed.

**Reporting summary.** Further information on research design is available in the Nature Portfolio Reporting Summary linked to this article.

## Data availability

Data supporting the findings of this work are available within the paper and in the Supplementary files. Source data for figures and tables is included in the Supplementary Data. The data sets generated and analyzed for this study are available from the corresponding author upon request.

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

## Acknowledgements

This study is funded by the Undergraduate Research Program of China Agricultural University (X2022100190381) and the Research Program of the National Key Laboratory of Human Factors Engineering, Astronaut Center of China (HFNKL2023W08). We thank Professor Qimei Lin, College of Environmental Sciences of China Agricultural University for his support and recommendation, and his introduction of Associate-Professor Zhencai Sun as our supervisor and responding author. We also thank Professor Ronghui Pan and Hao Du, College of Agriculture and Biotechnology of Zhejiang University, for their permission to perform cultivation experiments in their lab. We thank Yike Zhang, an undergraduate student at China Agricultural University, for her support in the measurement of plant samples.

## Author contributions

Y.X. conceptualized the experiment and took the lead in writing the manuscript as a team leader. Y.Y. and C.L. developed the methodology and accomplished the culture experiment of PSBs. Y.X. carried out the cultivation experiment of plants. Z.S. guided all other authors to ensure the scientifity and strictness of the project and helped to improve the design of experiments as a supervisor. All authors (Y.X., Y.Y., C.L., and Z.S.) contributed to funding the project operations and writing the successful proposal. Y.X. accomplished the statistics and data visualization of the project. All authors contributed to the review and editing of the manuscript.

## Competing interests

The authors declare no competing interests.
