## [Peer Review File · Communications Biology]

Reviewers' comments:

Reviewer #1 (Remarks to the Author):

The main idea of this work is that phosphate-solubilizing bacteria help to solubilize the phosphorus in the lunar regolith and improve plant growth. In order to prove their point, authors chose five different bacteria and tested their phosphate solubilizing capacity in cultures in which the lunar regolith was added. Later, they attempted to use a combination of three different bacteria in the lunar regolith stimulant; and assess the impact of the presence of bacteria to plant health. Despite the idea that this work is novel and the conclusions would bring new perspectives to the growing field of Astro-botany, I think authors should address some important points.

MAIN POINTS:

- Despite the authors performing the phosphorus and growth analysis using single bacteria, the authors mixed bacteria for plant cultivation assay. We do not know if these different bacteria impact each other's growth and, more importantly, how they impact each other's phosphorus solubilizing capacity. Therefore, it is very hard to interpret the plant cultivation assay, which used three different bacteria. And if we look at the results, we can see that there is a negative impact of bacterial inoculation, particularly on Day 15. This suggests to me that either author should have inoculated the regolith stimulant with single bacteria for plant cultivation assay or measure the growth and performed phosphate solubilization assays of bacteria using the same three bacterial combination that they used for plant cultivation assays. This is important because we do not know how the community dynamics of the three bacteria change over time. Maybe one bacteria takes over 15 days later. Without assessing the dynamics of this small community of 3 bacteria, it is hard to know what is dead/what is alive by day 15 of inoculation; or how the community changes. Maybe the combination of three bacteria has toxic effects on plants and can change the growth environment.

-We also do not know how bacteria survive in the lunar regolith or how their abundance change in the lunar regolith. Although the authors show in Figure 2-3 that lunar regolith addition to bacterial culture does not have a toxic effect, we do not know the amount/dose of added lunar regolith. Plus, it is very different to add regolith to culture vs culture into regolith.

-Lastly, I can't entirely agree that the authors prove that bacteria improves plant performance (which is the main point of this work). Yes, we see that bacteria increase the phosphorus content on Day0,5 and 15. However, I would expect to see a difference if the phosphorus content of the plant on Day 5 and Day 15 if the bacteria were disassociating the phosphorus and giving it to the plant. Instead, we do not see any difference in the phosphorus content of plants between Day0 of inoculation with the bacteria and Day5. In other words, bacteria need time to disassociate the phosphorus. The effect we see here is most likely just due to the presence of bacteria, which can enrich the regolith and improve the plant's nutrient condition. If we had checked the nitrogen content of the plants, we could have seen an improvement in nitrogen, too, just due to the presence of bacteria. If the sterile control includes dead bacteria, this might change the interpretation of the data; but based on my understanding, it is just the inoculation medium (please clarify in the M&M).

Another point is that we do not see any difference in the biomass of plants between inoculated and non-inoculated plants. The data does not mean much since the day 5 plants are only two reps. The authors should have had more reps.

Other minor points:

Line 119 to 120: Authors test the impact of lunar regolith on the growth of different bacterial species and suggest that the toxic materials in regolith do not affect the growth of bacteria. This can be due to the dose of added regolith. How much regolith did the authors use in experiments described in Figure 2A. It should be indicated in the main text and the figure legend. Also, how did the authors decide the amount of regolith that is added to bacterial culture? Did they do a toxicity test?

Line 126-142: The same comment is valid here. Authors should indicate how much regolith they added and how they decided on that amount to see the impact of regolith on bacterial growth and PH

change in bacterial medium.

Figure 3: Authors should add the long names for treatments; it is very hard to understand the treatments by looking at the abbreviations in the legend.

Line 427-432: I'm afraid I have to disagree with the interpretation of this data: First of all, authors also look at the phosphorus content normalized with the biomass of the plant. If we look at that, the 15-day inoculation plants have a higher phosphorus content per milligrams of biomass. Authors should compare this with how much phosphorus Arabidopsis leaves contains typically. This would tell us whether the system pumped too much phosphorus to the plants, leading to high toxic levels. Secondly, the authors should have checked the pH of the regolith. This would tell us what happens to the regolith when bacteria process the phosphorus. Maybe too much processing of phosphorus in the regolith leads to the release of other toxic materials to the regolith, inhibiting plant growth.

Line 119 to 120: Authors test the impact of lunar regolith on the growth of different bacterial species and suggest that the toxic materials in regolith do not affect the growth of bacteria. This can be due to the dose of added regolith. How much regolith did the authors use in experiments described in Figure 2A. It should be indicated in the main text and the figure legend. Also, how did the authors decide the amount of regolith that is added to bacterial culture? Did they do a toxicity test?

Line 126-142: The same comment is valid here. Authors should indicate how much regolith they added and how they decided on that amount to see the impact of regolith on bacterial growth and pH change in bacterial medium.

Figure 3: Authors should add the long names for treatments; it is very hard to understand the treatments by looking at the abbreviations in the legend.

Line 427-432: I'm afraid I have to disagree with the interpretation of this data: First of all, authors also look at the phosphorus content normalized with the biomass of the plant. If we look at that, the 15-day inoculation plants have a higher phosphorus content per milligrams of biomass. Authors should compare this with how much phosphorus Arabidopsis leaves contains typically. This would tell us whether the system pumped too much phosphorus to the plants, leading to high toxic levels. Secondly, the authors should have checked the pH of the regolith. This would tell us what happens to the regolith when bacteria process the phosphorus. Maybe too much processing of phosphorus in the regolith leads to the release of other toxic materials to the regolith, inhibiting plant growth.

Reviewer #2 (Remarks to the Author):

The paper entitled: "Phosphorus-Solubilizing Bacteria Improved the Growth of Arabidopsis thaliana on Lunar Regolith Simulant by Dissociating Insoluble Inorganic Phosphorus" describe the findings of five phosphorus-solubilizing bacteria and how they might be used to remediate a version of the lunar regolith simulant CAS-1 to improve Arabidopsis thaliana growth.

The paper is not well written and was hard to follow. It is recommended that further copy editing, and sentence re-phrasing be done to improve the readability of the manuscript in order to understand the science represented.

There are several fundamental concepts that were not well discussed in the paper.

1. The used of a "copy" of the lunar regolith simulant, CAS-1, to perform such experiments is not clearly explained. Was the scoria collected by the authors subjected to the same XRF analysis performed by Zheng Y. et al. [80] group before used in this paper? Data used to describe the characteristic of the simulant were from Zheng Y. et al and not a comparison between the newly collected samples with the ones reported by Zheng in 2009. Also, the surface of the sampling site was covered with vegetation, this is a little concerning with regards to the microbiome and nutrients within the samples that could benefit plant growth in general. Further elaborations on the comparison between the newly collected samples and the previously cited nutrient analysis should be conducted and compared.

2. There was no discussion about the oxidative nature of the lunar regolith and how that might impact the application of the work demonstrated here?

3. Different phosphorus forms were characterized but methods and explanation how to distinguish between these categories of phosphorous detection was not thoroughly explained in the methods section.

a. Insoluble phosphorus,

b. Dissolved phosphorus, separated into two further categories- dissolved inorganic phosphorus (DIP) and dissolved organic phosphorus (DOP),

c. Microbial biomass phosphorus (MBP) also further separated into two categories: suspended microbial biomass phosphorus (SMBP) and attached microbial biomass phosphorus (AMBP).

d. Adsorbed phosphorus (AP)

4. It is recommended that an additional figure diagramming the workflow be incorporated to help the reader visualize your experimental setup and sampling procedures. As it is written, some areas were confusing and not clearly understood.

Ie: 1) "The simulant was remixed every 2-3 days" until ready for use? Further clarification is needed on the intent of this statement.

2) "The treated simulant was dried at 70°C to constant weight" will this not kill the inoculated microbes? Was this the intent or not? Clarification is required.

3) "and treated with the mixed diluent culture medium sterilized through filtration at 0 DAI and 21 DAI." The culture medium was sterilized. But did you not want the microbes present?

5. The replicates for the Arabidopsis experiment were insufficient to account for growth variations.

a. "In this study, we used four replicates for the culture of PSBs and five for the cultivation of Arabidopsis thaliana. However, due to pollution or other factors, some replicates were stopped from culture or cultivation during the culture. The decision was made on the status of each replicate, including color of the culture, spontaneous precipitation of cells, growth of plants or data from the measurement. We managed to keep at least two available replicates at the end of the experiment. The full description of replicate numbers (and corresponding data numbers used for analysis) at each time of sampling is provided in Supplementary Materials, Table 1."

So, in certain samples, only 2 replicates were used to perform the statistical analyses? For Arabidopsis samples, this is not enough. The paper by Paul et al had very limited lunar regolith to perform the experiments and thus loss of samples were justified in the report. However, performing an experiment with simulant should be more thoroughly conducted with sufficient replicates as assess to simulants should not be as limited.

6. In terms of the results generated. Figure 5d should also include the horticultural soil samples for comparison. Also, the conclusion of the results in figure 5 was not well justified nor explained. At 0 DAI the Arabidopsis plants grew the best – without any time to amend the simulant yet the conclusion is that the PSB is needed to dissociate the simulant? Especially, since in Figure 3, the graphs show at least a few days needed to dissolve the phosphorus. It suggests that something else within the culture medium (nutrients) that is enabling the plants to grow instead of the PSB augmenting the simulant.

Reviewer #3 (Remarks to the Author):

The research Ms entitled: Phosphorus-Solubilizing Bacteria Improved the Growth of Arabidopsis thaliana on Lunar Regolith Simulant by Dissociating Insoluble Inorganic Phosphorus, provides new findings regarding the application of PSB on a simulant of lunar regolith with the assumption that can simulate a future scenario for improving lunar regolith and P uptake using PSB for cultivation of higher

plants in future lunar bases. Based on the literature, this is an original work that presents original data; however, the most important question is, how realistic is the hypothesis addressed in this Ms to appropriately speculate the need for using lunar regolith (poorer not only of P) for crop farming purposes in the future (and when?). Although the Ms is well presented and results are promising, attention should be paid throughout most Ms's sections. Here below a few comments:

Abstract:

Only 3 sentences were devoted to highlight the key findings, however authors are invited to provide the most attractive part of the results that addresses the content of the Ms and the title.

In "...dissolved salinity from the regolith simulant, and some of the phosphorus-solubilizing bacteria can dissociate the insoluble phosphorus in the regolith simulant." : Is THIS P NATIVELY EXISTS IN regolith or there was a TCP ADDED TO THE CULTURE?

Introduction:

Comment on Fig.1:

COMPOSITION (instead of composition)

CAS-1 and A14 need to be identified

Fig C is confusing at certain level/

FIRST: colors are confusing with the same colors used in Fig b:

In fig c, It is not clear why do bars in the before culture treatment present different P concentrations? Shouldn't the concentration be the same and at a minimal level since no solubilization is expected at this stage.

Results and discussion:

THE FIRST COMMENT is about the ability of PSB TO SOLUBILIZE P, throughout the MS, AUTHORS SOULD MAKE CLEAR THE ORIGIN OF P SOLUBILIZED, if IS IS FROM TCP OR from regolith, please make sure you report this clearly to avoid confusion.

In "smaller than.." Smaller or bigger? In section 1, it is mentioned: "Therefore, the particle size of the simulant used in this study was 0.3-0.5mm".

In "...verified in the previous $\text{Ca}_3(\text{PO}_4)_2$ decomposition.... in the simulant: This is Solubilization of tricalcium P ADDED TO SIMULANT, my question here is what about the fraction of P Natively existing in simulant (0.4 – 0.8 % in fig. 1B), did the author tested whether inoculated simulant (without CaP) with PSBs would solubilize some P

Fig 2: In figure 2C: authors would have measured soluble P content among the total soluble salt;

In section: "Dynamics of phosphorus ... treated by PSBs.":

IN THIS SECTION, FROM L1 to L15: this needs to be displaced to methodology given that authors describes the different forms of P measured rather than describing or interpreting results.

In addition, supporting references are worth citing

Is "classification" the right word to use?

My question here, why haven't you measured the PSB effect on regolith to dissociate the native P in it?

Could you please specify the origin of the P dissolved in culture, medium? Was it TCP or regolith?

THIS SOULD BE "adsorbed or immobilized" NOT "absorbed"

Are you sure sodium CARBONATE will desorb P from regolith?, unless you provide a reference;

In "system in the form of dissolved phosphorus and MBP, and..." : This is confusing, This is not understood, how and why P form added the culture medium was dissolved P and MBP?

Legend of Fig. 3: As a reviewer it took me enough time to understand the figure: 1) many subfigures, 2) many P forms presented, 3) P forms should be in each sub-figure so there is no need to check in the legend, 4) the legend should be informative than in its present form, the abbreviation DIP, DOP, should be spelt out for a more clarity 5) legend should also describe what the control is as to make it easy for the reader to analyse, here control values, in a, b,c are quite high, why? Even though lower than treatments but still around 200 – 250 ppm (MBP, DIP), etc...

In "...adsorption and combination while...": what do you mean by combination?
In "be absorbed by higher plants." : OR USED BY SOIL MICROBES

In "Adsorbed phosphorus is a potential source of phosphorus" : this sentence is not understood

In section: The mechanisms of the microbial dissociation: the major comment is
That the section examines mechanisms of MICROBIAL P solubilization rather than describing available P dissolved, mechanisms mean strategies involved to make P available, i.e., organic acids, enzymes, molecules, siderophores, etc...

In "simulant, we defined the concept of total available phosphorus (TAP) which included all forms of phosphorus, except...: HOW DOES IT VARY with existing concept or methodology?

In " the sum of initial exogenous phosphorus input and all phosphorus dissociated by PSBs. : This is not clear? What was that exogenous P, in what form, etc.

In " that there was little dissolved phosphorus in the lunar regolith simulant before the culture,: That was one of the issues I am still highlighting.

In " entirely provided by liquid medium..": Weren't you able to provide liquid medium with no P available? How does the inoculant solution provide P before culture?

In " This showed that B. mucilagenosus had a strong ability to excrete organic matter that .." : WAS this measured?

In "...the contribution of adsorbed phosphorus to...": Did authors mean the adsorbed P (insoluble) in bacterial Cells?

In section "Plant growth in the lunar surface regolith simulant treated with PSBs": L1 - 4 all are Methodology not results.

In "Neither treatment hadgrowth.: THE sentence is not complete

In ". The plants showed severeat 21 DAG. / this NEEDS TO BE EXPLAINED:rewritten

In " The results suggested that the ...the simulant. : how,?

In "...The higher phosphorus content indicated ... phosphorus elements. : Why?

In "...which led to the increase of soluble salt content in the soil, resulting in severe salt stress on plants and inhibiting the further growth. : Do you think % soluble salt around 3-4% after 250 h of culture will inhibit plant growth? This is actually what authors measured in vitro, but should this be extrapolated to the in plant experiment as an explanation to the growth inhibition?

In " Meanwhile, the experiment revealed the negative impacts caused by the prolonged treatment of PSBs, which may be caused by high soluble salt content.: THIS is still a speculative statement given that there is no evidence provided here to support quantitatively this conclusion.

In Methods

In "It should be noted that the culture medium for B. mucilagenosus was silicate bacteria medium (glucose 5 g/L, MgSO₄ 0.5g /L, CaSO₄·2H₂O 0.1 g/L, Na₂HPO₄ 2 g/L, FeCl₃ 0.005 g/L, pH=7.0),: authors needs to justify why?

In provided by our laboratory.: Please define the laboratory name

In: The culture medium was sampled before and after culture, and..." was this AFTER INOCULATION?

In: shaking flask culture experiment of PSBs with lunar regolith simulant: please improve the title

In: "...and the possibility of microbial dissociation on insoluble inorganic phosphorus by PSBs,...": here it is not clear what the author wanted to explain, as it is the statement is similar to the most literature about PSB aiming to solubilize insoluble P. Is the purpose here to solubilize P FROM REGOLITH, or add tricalcium P mixed it with regolith and quantify THE SOLUBLE ?

In: . "The solid phase consisted of 40 g of the lunar surface regolith simulant which was pre-dried to constant weight at 70°C in a 150 ml flask.." : GRANULOMETRIE?

In: "...40g of solid phase within an ultra-clean workbench, and sealed a sealing film of air-permeability. Before inoculation, the mixed culture medium was cultured at 30°C and 180 rpm for 24 hours to check whether it was contaminated. : Authors needs to explain using 40g of simulant, and why not testing different amount before choosing only one for the rest of experiments (here 40g)?

In : 6. Sampling : TITLE NEEDS TO BE INFORMATIVE

In: "The seeds used in the experiment included lettuce (*Lactuca sativa*) and common wheat (*Triticum aestivum*). According to the shaking flask culture experiment results, the tested PSBs were determined to be *B. mucilaginosus*, *B. megaterium*, and *P. fluorescens*. : Section 7 lacks explanation about the purpose of the seed germination experiment

In: The culture medium of the above three PSBs in the regolith simulant culture experiment at 0 and 21 DAI: not clear why 0 and 21?

In: "was fully moistened with diluted culture medium or sterile water.: Thea reason why inoculation was done 0 & 21 DAI and using diluted PSB-rigolith culture is not clear In addition, it is questionable why *Arabidopsis* seeds were not tested for germination potential and rate along with wheat and lettuce, particularly *Arabidopsis* stands as the main tested plant species in this research.

In: . To stop culture, five bottles...": WHAT DOES IT MEAN BY "TO STOP CULTURE" and what is the purpose of this? My question here is, why *arabidopsis* growth was not tested under simultaneous inoculation and addition of simulant (this would have taken as a treatment), instead of adding a simulant that was previously inoculated and incubated then used to grow *Arabidopsis*?

In: "...;0.2 ml 1/4 MS medium: WHAT IS MS medium?

In: "However, due to pollution or other factors, some replicates were stopped from culture or cultivation during the culture.: In this case, how is it likely possible to statistically interpret results of treatments with less than 3 replicates according to the Supplementary Table 1.

Dear reviewers,

We deeply appreciate the careful examination of our manuscript from all three reviewers, and we wrote this responding document to you with all of our respect. In this document, we listed each of the three reviewers' comments in turn in a table, gave point-to-point responses, and refined our manuscript to the best of our ability. We sincerely hope that these responses and changes could address every mistakes, issues and misunderstandings in our manuscript, and the manuscript could clearly express the science it represents to your satisfaction. We are expecting to your approval of this manuscript, and we really look forward to seeing it accepted in the near future.

We would like to thank you with great gratitude.

If there is any questions, please contact us.

Authors

The order of the tables:

Table 1, Critical comments and revisions;

Table 2, Comments from reviewer #1;

Table 3, Comments from reviewer #2;

Table 4, Comments from reviewer #3.

Table 1. Critical comments and revisions;

No.	Critical comments	Author Respond	Change made
1	Appropriate controls must be added to exclude the possibility that the inoculation medium (rather than solubilization of phosphorous by the inoculum) enhances plant performance (requested by Reviewer #1)	We are grateful to the reviewers for highlighting this critical issue. Although the controls were correctly set up in the first submission, as we wrote in the Methods, section 8, "In sterile control, 2 ml glucose broth medium and 0.2 ml mixed steam-sterilized inoculation solution were added", unfortunately the ambiguity of our wording prevented the reader from correctly understanding our intention. We realized that the name of this control "sterile" would be easily misinterpreted that no PSBs was added to this group, instead of adding PSBs and then killed them in steam-sterilize procedure to be "sterile". Meanwhile, the description of "inoculation solution" was not clear. Although it was described in Methods, section 2, it is still difficult to make connections between the two. To make our descriptions clear, we re-named our controls and treatment groups. The "Sterile" control was renamed as "Sterilized" control, as we hope that this could minimum the misunderstanding. We also added a paragraph in Method, section 7(the former section 8), to describe how we prepared the inoculation solution in detail, and mentioned how we had used this concept in section 2.	1. Rename the "Sterile control" as "Sterilized control". 2. In Methods, section 7, we added a new paragraph after the first graph, describing the preparation of the "sterilized control". "We start the preparation from.....for subsequent experiments"
2	Demonstrate that the inoculum can solubilize phosphorous that natively exists in the lunar regolith (requested by Reviewer #3).	We are sorry that our description was so unclear that the reviewer #3 did not understand our experiment. Seemingly our description about "Standard test for the ability of microbes to dissolve insoluble inorganic phosphorus", in which TCP was added, led to a misunderstanding that TCP was also added to the lunar regolith simulant. As a matter of fact, the culture medium itself could not leach native phosphorus from the simulant, as we had shown in fig. 2 a-c and explained in the last paragraph of the section named as "Dynamics of.....by PSBs" in Results & Discussion. The increase of total available phosphorus content that we had observed in the shaking flask experiment should be completely contributed by PSBs treatments. To make our descriptions clear, we adjust the order of related narration in the manuscript. We also added a new sentence that could reduce the misunderstanding, which is given on the right.	1. The first sentence of the Results & Discussion, paragraph 2, "To study the interactions.....living conditions for microorganisms" was removed to paragraph 3. 2. At the end of the paragraph 2, "in the simulant" was replaced by "from Ca ₃ (PO ₄) ₂ ". Then, a new sentence was added as "However,.....remains to be further verified."
3	More replicates must be performed for the Arabidopsis experiment to account for growth variation and enable reliable statistical analysis to be performed on the data obtained from this experiment.	We are sorry that we mistakenly assumed that the small number of replicates would be sufficient to support our conclusions, when writing the first version of the manuscript. Thanks to the criticisms from three reviewers, we are well aware that the number of replicates is needed to be more. Thus, we re-designed our cultivation experiment, in which more replicates were performed and the preparation method was improved to reduce the sample loss. Detailed adjustments can be found on the right.	Each treatment or controls contained 12 replicates, and most of them were available during the whole cultivation period(only one replicate, the C1 of the Sterilized control group, did not provide any surviving plants at 24 DAS). The adjustments on the preparation of cultivation substrate included the following main points: 1) The amount of glucose broth medium and inoculation solution was half of the original, and the remaining volume was supplemented by ultra-pure water; 2) The plant material was replaced by Nicotiana benthamiana , which could grow more robust than Arabidopsis thaliana , and being less likely to die in bad conditions. 3) More treatment groups were settled, from the 0, 5, 15 DAI treatments to "Not pre-cultured" and pre-cultured for 6, 12, 18 days. 4) More plants in each replicate were retained. The possibility of interfering results by thinning seedlings

			subjectively would therefore be reduced.
4	The presentation of the manuscript should be improved based on the suggestions provided by Reviewers #2 and #3.	We thank reviewers for providing constructive suggestions to improve our manuscript. We have thoroughly and carefully examined our manuscript and revised the article in accordance with all the reviewer's suggestions.	The detailed changes on presentation of the manuscript are listed in the table in response to specific comments from individual reviewers.
5	Additional methodological information needs to be provided, particularly regarding how the lunar regolith simulant used in the study was determined to be similar to CAS-1 (requested by Reviewer #2).	We understand the importance of the elemental composition of our lunar regolith simulant on the conclusions in our manuscript. Therefore, we have tested the elemental composition before our first submission. Unfortunately, the results were not provided in time that we could not put them in the manuscript. Now we would like to add these information to our manuscript, including a new figure and supplementary methodological information. We are satisfied that the additional information do not change the conclusions in our manuscript.	1. Fig. 1 b has been reworked to include information about the elemental composition of the sample used in the experiment. 2. The measurement method of the elementary composition of samples is provided in the last paragraph of Methods, section 1.

Table 2. Comments from reviewer #1.

No.	Comments From Reviewer #1	Author Respond	Change made
1	- Despite the authors performing the phosphorus and growth analysis using single bacteria, the authors mixed bacteria for plant cultivation assay. We do not know if these different bacteria impact each other's growth and, more importantly, how they impact each other's phosphorus solubilizing capacity. Therefore, it is very hard to interpret the plant cultivation assay, which used three different bacteria. And if we look at the results, we can see that there is a negative impact of bacterial inoculation, particularly on Day 15. This suggests to me that either author should have inoculated the regolith stimulant with single bacteria for plant cultivation assay or measure the growth and performed phosphate solubilization assays of bacteria using the same three bacterial combination that they used for plant cultivation assays. This is important because we do not know how the community dynamics of the three bacteria change over time. Maybe one bacteria takes over 15 days later. Without assessing the dynamics of this small community of 3 bacteria, it is hard to know what is dead/what is alive by day 15 of inoculation; or how the community changes. Maybe the combination of three bacteria has toxic effects on plants and can change the growth environment.	We completely agree with the comment on interactions between three tested strains. However, we are sorry that the fundings of our research program was so limited that only a simple and primary cultivation experiment could be carried out with three PSBs added to the same ratio. In fact, it is inspiring that the results of such a primary experiment could confirm the possibility that we transform lunar regolith into cultivation substrate with PSBs treatment. We are truly looking forward to carrying out further experiments to find out the community dynamics of the PSBs, and figure out the main effecting PSBs strains in different culture stages, the best inoculating ratio of three PSBs, etc. However, none of these experiment would not be realistic before the acceptance of this manuscript and our success in applying for more fundings.	No change has been made for this comment.
2	-We also do not know how bacteria survive in the lunar regolith or how their abundance change in the lunar regolith. Although the authors show in Figure 2-3 that lunar regolith addition to bacterial culture does not have a toxic effect, we do not know the amount/dose of added lunar regolith. Plus, it is very different to add regolith to culture vs culture into regolith.	Thank you for your comments. We agree with you that the abundance of PSBs should be examined, therefore we have measured the OD600 of the soil extract solution, as the growth index of PSBs in the simulant during the cultivation of plants. The ratio of simulant and LB medium is added to the main text and the explanatory text of fig. 2a, according to your request. We agree that it is different to add regolith into the culture vs the opposite. In fact, two types of experiment are both applied to our experiments (shaking flask experiment and cultivation experiment). Due to the limited fundings, we had to make a risky assumption that the same procedure of dissociating phosphorus would happen in both experiment. Luckily, the outcomes of the both experiments proved that we were right to some extent, and the PSBs that was active in the flask experiment did improved the growth of plants in the regolith.	1. The OD600 of soil extract solution from soil samples in the cultivation experiment was measured and provided in fig. 5e. 2. The legend of fig. 2a was added with the sentence, "The mass ratio of regolith simulant and LB medium was 1:2."
3	-Lastly, I can't entirely agree that the authors prove that bacteria improves plant performance (which is the main point of this work). Yes, we see that bacteria increase the phosphorus content on Day0,5 and 15. However, I would expect to see a difference if the phosphorus content of the plant on Day 5 and Day 15 if the bacteria were disassociating the phosphorus and giving it to the plant. Instead, we do not see any difference in the phosphorus content of plants between Day0 of inoculation with the bacteria and Day5. In other words, bacteria need time to disassociate the phosphorus. The effect we see here is most likely just due to the presence of bacteria, which can enrich the regolith and improve the plant's nutrient condition. If we had checked the nitrogen content of the plants, we could have seen an improvement in nitrogen, too, just due to the presence of bacteria. If the sterile control includes dead bacteria, this might change the interpretation of the data; but based on my understanding, it is just the inoculation medium (please clarify in the M&M).	Thank you for your comment. As a critical comment listed in the table 1, we have placed the response to this comment in the row No.1 of table 1.	Please check the row No.1 of table1 for corresponding changes.
4	Another point is that we do not see any difference in the biomass of plants between inoculated and non-inoculated plants. The data does not mean much since the day 5 plants are only two reps. The authors should have had more reps.	This comment is also listed as critical in the table 1. We have placed the response to this comment in the row No.3 of table1.	Please check the row No.3 of table 1 for corresponding changes.
5	Line 119 to 120: Authors test the impact of lunar regolith on the growth of different bacterial species and suggest that the toxic materials in regolith do not affect the growth of bacteria. This can be due to the dose of added regolith. How much regolith did the authors use in experiments described in Figure 2A. It should be indicated in the main text and the figure legend. Also, how did the authors decide the amount of regolith that is added to bacterial culture? Did they do a toxicity test?	Thank you for this comment. As we have mentioned in row No.2, Supplementary description about the ratio of the regolith and LB medium has been added to the manuscript. As a matter of fact, we have conducted a series of pre-experiments with different amount of regolith added to the culture, including half of the weight of the culture to ten times of the weight of the culture. We observed that: when the amount of regolith was more than the culture, shaking the culture to maximum the microbial exposure to the oxygen	Descriptions was added as described in row No.2.

		was impossible. Manual stirring of the culture may introduce subjective errors, like the time or the degree of the stirring procedure. An extreme is adding culture to the regolith to a water ratio of ~20%. The moisture in the regolith allows the regolith to form a relatively loose structure, which is conducive to microbial respiration. However, fewer culture medium prevented microorganisms from further growing, and the regolith treatment was less adequate, as we have observed in the preparation of cultivation substrate. The other extreme is adding a small amount of regolith to the culture, which was described in reference 39 & 41. We believed that such treatments are less practicle to be performed on the moon, as water is very rare and valuble on the moon. Thus, we finally chose the ratio of 1:2, which enables the microbe to be shaking-flask cultured and also a more adequate treatment of regolith by PSBs. We did not perform toxicity test, as we read from reference 39 & 41 that untreated lunar regolith simulant are "inert" and less likely to affect biological activities, neither positive nor negative. In fact, we observed from our experiment that the regolith simulant treated with different PSBs may show different impact on microbial growth (salt content, pH), at a addition ratio of 1:2. We may study this toxicity more systematically in the future.	
6	Line 126-142: The same comment is valid here. Authors should indicate how much regolith they added and how they decided on that amount to see the impact of regolith on bacterial growth and PH change in bacterial medium.	The response above is also valid for this comment.	Descriptions was added as described in row No.2, table 2.
7	Figure 3: Authors should add the long names for treatments; it is very hard to understand the treatments by looking at the abbreviations in the legend.	Thank you for this comment. We have adjusted the legends in the Fig. 3 and other figures, using full name for all treatments instead of abbreviations. We also made the title of each subfigure more informative.	Most of the possible confusing abbreviations in the manuscript have been changed to their full names, including DOP, DIP, MBP, AMBP, SMBP, TAP. Other important abbreviations are remained, including DAS, DAI, PSB, GIR and DIIP.
8	Line 427-432: I'm afraid I have to disagree with the interpretation of this data: First of all, authors also look at the phosphorus content normalized with the biomass of the plant. If we look at that, the 15-day inoculation plants have a higher phosphorus content per milligrams of biomass. Authors should compare this with how much phosphorus Arabidopsis leaves contains typically. This would tell us whether the system pumped too much phosphorus to the plants, leading to high toxic levels. Secondly, the authors should have checked the ph of the regolith. This would tell us what happens to the regolith when bacteria process the phosphorus. Maybe too much processing if phosphorus in the regolith leads to the release of other toxic materials to the regolith, inhibiting plant growth.	We sincerely thank you for your careful examination of our manuscript, and we agree that the risk of toxicity caused by high P level could not be excluded if the comparison with typical content of p was missing. In our supplementary experiment, we did setted a group of plants that were grown in horticultural soil. Unfortunately, Due to the limit of very small weight of plant samples, we could not measure the phosphorus content in plant samples. However, given the data on other indexes that we had examined, we are still confidient to say that the treatment of PSBs did improved the growth of plnats in regolith simulant. We are also glad to find out that the inhibitory impact that 15 DAI treatment had on plant growth dissappeared in our new cultivation experiment, after we adjusted our preparation method of cultivation substrate. We assumed that the high P level and relatively low salt content (compared with former 15 DAI treatment group) may contributed to the better growth of plants.	No change has been made for this comment.

Table 3. Comments from reviewer #2.

No.	Comments From Reviewer #2	Author Respond	Change made
1	The paper is not well written and was hard to follow. It is recommended that further copy editing, and sentence re-phasing be done to improve the readability of the manuscript in order to understand the science represented.	We appreciate your criticism of the language issues in our manuscript. We the authors have conducted several meetings and discussions to perfect the language in our manuscript according to the comments from three reviewers. We have also sent our manuscript out to our colleagues (they claimed to be anonymous) who are experts in english writting for additional advices. We sinsecely hope that the revised manuscript could be better written and easier to be understood.	No particular change is made for this comment. Other changes related to the language issues could be found above and below.
2	1. The used of a "copy" of the lunar regolith simulant, CAS-1, to perform such experiments is not clearly explained. Was the scoria collected by the authors subjected to the same XRF analysis performed by Zheng Y. et al. [80] group before used in this paper? Data used to describe the characteristic of the simulant were from Zheng Y. et al and not a comparison between the newly collected samples with the ones reported by Zheng in 2009. Also, the surface of the sampling site was covered with vegetation, this is a little concerning with regards to the microbiome and nutrients within the samples that could benefit plant growth in general. Further elaborations on the comparison between the newly collected samples and the previously cited nutrient analysis should be conducted and compared.	Thank you for your comment. As a critical comment listed in the table 1, we have placed the response to this comment in the row No.5 of table 1. In addition, we did not perform XRF analysis for our samples. According to our understanding, XRF is a method to detect the elemental composition of the solid samples. Unfortunately, our limited fundings prevented us from conducting such experiments. Alternatively, we used the atomic flame spectrophotometer to analysis the composition of several elements, including Fe, Ca, Mg, Na, K, P, and Mn. The results are very similar to CAS-1, as well as the samples from Apollo 14 mission. We believe that it is sufficient to support the conclusions in our experiments.	Please check the row No.5 of table 1 for corresponding changes.
3	2. There was no discussion about the oxidative nature of the lunar regolith and how that might impact the application of the work demonstrated here?	Thank you for your comment. We agree with you that the additional analysis on the oxidative nature of the lunar regolith would better support our conclusions, especially on how the regolith inhibited the	No particular change is made for this comment.

		growth of microbes. However, we believed that this is not a part of the core content of our manuscript, in which the most important part is to find out that PSBs treatment promoted the growth of plants in lunar regolith simulant. We are sorry to say that no related analysis was done due to the limited fundings. However, we are ready to supplement experiments on the oxidative nature of the lunar regolith when funds are available.	
4	3. Different phosphorus forms were characterized but methods and explanation how to distinguish between these categories of phosphorous detection was not thoroughly explained in the methods section.	We regret that the vague description in our manuscript prevented the reviewers from finding the description of how the various phosphorus forms were determined. As a matter of fact, we have placed this part of measurement methods in the Supplementary Materials. We believe that it would be better to leave these methodological descriptions in Supplementary Materials than putting them in main text, as they are too long and complicated. However, we are still willing to change the descriptions in the Method section of the main text to show that we have moved related part of descriptions to Supplementary Materials, so that the readers would be easily guided to them.	The description as follows is added to the end of the fifth step of section 6, Methods: "The detailed methods of laboratory measurements, including the measurement of dissolved inorganic phosphorus, dissolved organic phosphorus, microbial biomass phosphorus, suspended microbial biomass phosphorus, attached microbial biomass phosphorus and adsorbed phosphorus of five PSB treatments, are provided in Supplementary Materials, Measurement Methods."
5	4. It is recommended that an additional figure diagraming the workflow be incorporated to help the reader visualize your experimental setup and sampling procedures. As it is written, some areas were confusing and not clearly understood. le: 1) "The simulant was remixed every 2-3 days" until ready for use? Further clarification is needed on the intent of this statement. 2) "The treated simulant was dried at 70°C to constant weight" will this not kill the inoculated microbes? Was this the intent or not? Clarification is required. 3) "and treated with the mixed diluent culture medium sterilized through filtration at 0 DAI and 21 DAI." The culture medium was sterilized. But did you not want the microbes present?	We are grateful for your constructive comments, and we reworked on our figures to insert a visual description of the cultivation and sampling workflow (Fig. 5a). We hope that this could make our descriptions more understandable. Responses to subcomments: 1) We are sorry that this caused confusion. The aim of this step was to maximum the microbial exposure to the oxygen, so that the PSBs could respire normally. We have added the related explanation to the manuscript. 2) The original purpose of this step was to stop the microbial activities in the regolith, and minimum the influence on the chemical properties of the regolith (otherwise higher temperature would be used). In our new supplementary cultivation experiment, we canceled this procedure to allow PSBs continue to grow after sowing. Thus, related description was deleted from our manuscript. 3) Thank you for asking clarification. Indeed, we did not hope that the microbes would have direct interactions with the seeds, as we had observed in pre-experiments that the microbial exposure to the seeds would lead to infection and the seedlings would stop growing and die. However, in our new experiment, the germination experiment lost its meaning, and was deleted from the main text.	1. Fig. 5a is added to show the workflow of cultivation and sampling procedure. 2. The descriptions in section 7, Methods, is replaced by "The simulant was remixed every 2-3 days, so that the phosphorus solubilizing bacteria are fully exposed to oxygen for normal respiration". 3. The step of heating for 70°C in the preparation of cultivation substrate is deleted. 4. The figure (Fig. 5a in the version of first submission) and descriptions related to germination test are deleted from the manuscript.
6	5. The replicates for the Arabidopsis experiment were insufficient to account for growth variations. a. "In this study, we used four replicates for the culture of PSBs and five for the cultivation of Arabidopsis thaliana. However, due to pollution or other factors, some replicates were stopped from culture or cultivation during the culture. The decision was made on the status of each replicate, including color of the culture, spontaneous precipitation of cells, growth of plants or data from the measurement. We managed to keep at least two available replicates at the end of the experiment. The full description of replicate numbers (and corresponding data numbers used for analysis) at each time of sampling is provided in Supplementary Materials, Table 1." So, in certain samples, only 2 replicates were used to perform the statistical analyses? For Arabidopsis samples, this is not enough. The paper by Paul et al had very limited lunar regolith to perform the experiments and thus loss of samples were justified in the report. However, performing an experiment with simulant should be more thoroughly conducted with sufficient replicates as assess to simulants should not be as limited.	Thank you for this comment. This comment is also listed as critical in the table 1. We have placed the response to this comment in the row No.3 of table 1.	Please check the row No.3 of table 1 for corresponding changes.
7	6. In terms of the results generated. Figure 5d should also include the horticultural soil samples for comparison. Also, the conclusion of the results in figure 5 was not well justified nor explained. At 0 DAI the Arabidopsis plants grew the best – without any time to amend the simulant yet the conclusion is that the PSB is needed to dissociate the simulant? Especially, since in Figure 3, the graphs show at least a few days needed to dissolve the phosphorus. It suggests that something else within the culture medium (nutrients) that is enabling the plants to grow instead of the PSB augmenting the simulant.	We sincerely thank you for your careful examination of our manuscript, and we agree that horticultural soil samples for comparison would be needed. In our supplementary experiment, we did setted a group of plants that were grown in horticultural soil, and compared the growth of plants with other treatments. In addition, we did not observe the same phenomenon of inhibition of plant growth in the "Pre-cultured for 18 days" as we had observed in the former 15 DAI group. We believe that this phenomenon may serve as a result of small replicate numbers, or inappropriate experimental operations, which is less credible.	The "Horticultural soil" group of plants has been added to our supplementary cultivation experiment.

Table 4. Comments from reviewer #3.

No.	Comments From Reviewer #3	Author Respond	Change made
1	Only 3 sentences were devoted to highlight the key findings, however authors are invited to provide the most attractive part of the results that addresses the content of the Ms and the title.	We sincerely appreciate you for your constructive comments on our abstract. Indeed, we realized that the abstract of our manuscript in the first submitted version was not adequate to represent the highlights of our study. Therefore, we worked to rearrange the sentences in the abstract, which could be found on the right. We hope that the new version of abstract could better represent our key findings.	The new version of abstract is as follows: In-situ utilization of lunar soil resources will effectively improve the self-sufficiency of bioregenerative life support systems for future lunar bases. Therefore, we explored the microbiological method to transform lunar soil into a substrate for plant cultivation. In this study, five species of phosphorus-solubilizing bacteria were used as test strains, and a 21-day bio-improving experiment with another 24-day Nicotiana benthamiana cultivation experiment was carried out on lunar regolith simulant. We observed that the phosphorus-solubilizing bacteria Bacillus mucilaginosus , Bacillus megaterium , and Pseudomonas fluorescens could tolerate the lunar regolith simulant conditions and dissociate the insoluble phosphorus from the regolith simulant. The phosphorus-solubilizing bacteria treatment had improved the available phosphorus content of the regolith simulant, promoting the growth of Nicotiana benthamiana . Here we proved that the phosphorus-solubilizing bacteria could effectively improve the fertility of lunar regolith simulant, making it a good cultivation substrate for higher plants. The results can lay a technical foundation for plant cultivation based on lunar regolith resources in future lunar bases.
2	In "...dissolved salinity from the regolith simulant, and some of the phosphorus-solubilizing bacteria can dissociate the insoluble phosphorus in the regolith simulant." : Is THIS P NATIVELY EXISTS IN regolith or there was a TCP ADDED TO THE CULTURE?	We thank you for your comment on our unclear wording. As a matter of fact, no TCP was added to regolith. As a critical comment listed in the table 1, we have placed the detailed response to this comment in the row No.2 of table 1. In addition, we replaced the expression "dissociate the insoluble phosphorus in the regolith simulant" with "dissociate the insoluble phosphorus from the regolith simulant". We hope that these adjustments could reduce the misunderstanding.	
3	COMPOSITION (instead of composition)	We are terribly sorry for our misspelling. We have replaced the misspelled words with right ones, and checked the whole manuscript to minimum misspelled words.	"composition" was replaced by "composition".
4	CAS-1 and A14 need to be identified	We are sorry for the missing of identification in the legend. We have added the description about CAS-1 in the legend and replaced "A14" with "Apollo 14 samples" which is more informative.	1. "CAS-1" in the legend of fig. 1 is replaced by "CAS-1 lunar regolith simulant", and added reference. 2. "A14" in the legend of fig. 1b is replaced by "Apollo 14 samples".
5	colors are confusing with the same colors used in Fig b:	Thank you for your comment. We have changed the colors of fig. 1c from orange and green to purple and yellow, according to your comment.	The colors of fig. 1c have been changed from orange and green to purple and yellow.
6	In fig c, It is not clear why do bars in the before culture treatment present different P concentrations? Shouldn't the concentration be the same and at a minimal level since no solubilization is expected at this stage.	Thank you for your comment. As a matter of fact, when inoculating the culture with PSBs, the inoculation solution would introduce a small amount of phosphorus to the system. It is easy to understand that the inoculation solution of different strains would contain different level of phosphorus concentration, as different strains would have different properties to absorb and release phosphorus. If we ignore this part of phosphorus addition, we may falsely consider this part of phosphorus as the phosphorus dissociated by PSBs, leading to false conclusions. Still, we would like to add description to clarify this, as we have shown on the right.	The following sentence is added to the legend of fig. 1c: "Different species of inoculated bacteria contain different amounts of phosphorus, which leads to different levels of phosphorus at the beginning of the test. "
7	THE FIRST COMMENT is about the ability of PSB TO SOLUBILIZE P, throughout the MS, AUTHORS SOULD MAKE CLEAR THE ORIGIN OF P SOLUBILIZED, if IS IS FROM TCP OR from regolith, please make sure you report this clearly to avoid confusion.	Thank you for your comment. As we have shown in No.2, table 1 and No.2, table 4, we have clarified our expressions to show that the origin of phosphorus is from the regolith instead of TCP, which was not added to regolith.	No particular change has been made for this comment.
8	In "smaller than..." Smaller or bigger? In section 1, it is mentioned: "Therefore, the particle size of the simulant used in this study was 0.3-0.5mm".	Thank you for pointing out this mistake. We have deleted the confusing expressions from the manuscript. The actual particle size that we used in our experiments was 0.3-0.5 mm, and we made it clear in our manuscript.	"we removed the part of the simulant with a particle size smaller than 0.3mm." is deleted.
9	In "...verified in the previous Ca ₃ (PO ₄) ₂ decomposition.... in the simulant: This is Solubilization of tricalcium P ADDED TO SIMULANT, my question here is what about the fraction of P Natively existing in simulant (0.4 – 0.8 % in fig. 1B), did the author tested whether inoculated simulant (without CaP) with PSBs would solubilize some P	Thank you for your comment and the response in No.2, table 1 is also valid for this comment.	No particular change has been made for this comment.
10	IN THIS SECTION, FROM L1 to L15: this needs to be displaced to methodology given that authors describes the different forms of P measured rather than describing or interpreting results.	We completely agree with you about this comment. Therefore, we moved the description about the classification of phosphorus to Supplementary Materials, section 2.	The changes have been made as described on the left.

	In addition, supporting references are worth citing		
11	Is "classification" the right word to use?	Thank you for this question. However, after a few rounds of discussion between authors, we believe that this word would be the suitable word to use. If you have any other suggestions, please let us know.	No particular change has been made for this comment.
12	My question here, why haven't you measured the PSB effect on regolith to dissociate the native P in it?	Thank you for your comment and the response in No.2, table 1 is also valid for this comment.	
13	THIS SHOULD BE "adsorbed or immobilized" NOT "absorbed"	Thank you for pointing out this mistake. We have made the correction according to your comment.	"absorbed" is replaced by "adsorbed".
14	Are you sure sodium CARBONATE will desorb P from regolith?, unless you provide a reference;	Thank you for your comment. As a matter of fact, using NaHCO ₃ solution to extract phosphorus from soil is a classical method in soil science, developed by Olsen, S. R. in 1951. We have described the method in Supplementary Materials, and provided the reference at the bottom. We also show you the reference here: Olsen, S. R. et al. Estimation of available phosphorus in soil by extraction with sodium bicarbonate. USDA Circular No. 939, US Government Printing Office, Washington DC.(1951)	Supplementary reference has been provided as shown on the left.
15	In "system in the form of dissolved phosphorus and MBP, and...": This is confusing, This is not understood, how and why P form added the culture medium was dissolved P and MBP?	Thank you for asking. To answer this question, we need to examine the composition of the glucose broth medium and the inoculation solution. In glucose broth medium, Yeast extract is used to provide essential nutrients and micronutrients, including phosphorus components, like phospholipid, phosphonucleotide, and Inorganic phosphate. These phosphorus components dissolve in the medium, contributing to the dissolved phosphorus. On the other hand, in the inoculation solution, there are living cells of the PSBs, in which holds the phosphorus components, contributing to the microbial biomass phosphorus. We hope that this explanation would help you to understand our experiments.	No particular change has been made for this comment.
16	Legend of Fig. 3: As a reviewer it took me enough time to understand the figure: 1: many subfigures, 2) many P forms presented, 3) P forms should be in each sub-figure so there is no need to check in the legend, 4) the legend should be informative than in its present form, the abbreviation DIP, DOP, should be spelt out for a more clarity 5) legend should also describe what the control is as to make it easy for the reader to analyse, here control values, in a, b,c are quite high, why? Even though lower than treatments but still around 200 – 250 ppm (MBP, DIP), etc...	Thank you for your constructive comments. We agree with you that the figure 3 was not well presented in the first submitted version. Therefore, we have reworked on the figure to make the titles and legends more informative. However, we failed to make the control of each legend more informative, due to the limit of space of each subfigure. Alternatively, we moved the descriptions of controls in section 5, Methods to the text of figure 3. We hope that these changes would make it easier to understand the figure. The reason why control groups have high level of dissolved phosphorus (inorganic & organic) is that the glucose broth medium contains such level of dissolved phosphorus, not from the leaching of regolith. Meanwhile, the content in the control group remained steady during the 21-day culture, indicating that the medium itself could not leach phosphorus from the regolith.	1. The vertical axis headings of each subfigure indicate the category of data that the figure represents. 2. All subfigures now have their own legend. 3. The following description is added to the text of figure 3: "In the control group of the phosphorus measurement, the medium mixed with simulant was not inoculated with PSBs. Each treatment included four flasks as replicates. The phosphorus content in the control group should be understood as the phosphorus contained in the glucose broth medium."
17	In "...adsorption and combination while...": what do you mean by combination?	Thank you for your careful examination of our manuscript. After checking, we realized that there are two mistakes in this expression. First, it should be "absorption" instead of "adsorption", to describe the progress that living cells uptake phosphorus contents in the environment. Second, the word "combination" means the combination between dissolved phosphorus components with the proteins and phospholipid on the surface of the cell. However, we did not find out related reference that could support this assumption. Therefore we deleted the word "combination".	"adsorption and combination" is replaced by "absorption".
18	In "be absorbed by higher plants.": OR USED BY SOIL MICROBES	We completely agree with your comment and we are sorry for the lapse in discussion. We have corrected our expression according to your comment.	"be absorbed by higher plants" is replaced by "be absorbed by higher plants or other microbes".
19	In "Adsorbed phosphorus is a potential source of phosphorus": this sentence is not understood	We are sorry that this sentence was not well-written. We have re-worked on this sentence to make it easier to be understood, and more related with the following descriptions.	" Adsorbed phosphorus is a potential source of phosphorus" is replaced by "Adsorbed phosphorus is a dynamic regulating reservoir of phosphorus elements."
20	That the section examines mechanisms of MICROBIAL P solubilization rather than describing available P dissolved, mechanisms mean strategies involved to make P available, i.e., organic acids, enzymes, molecules, siderophores, etc...	We agree with your comment for the description of total available phosphorus did not fit the title of this section. Thus, we moved the related discussion and fig. 4 to the section above.	The discussions about total available phosphorus, including two paragraphs and figure 4, has been moved from the section "The mechanisms of the microbial dissociation of phosphorus in the lunar regolith simulant" to the section "Dynamics of phosphorus content and forms in the lunar regolith simulant treated by PSBs".
21	In "simulant, we defined the concept of total available phosphorus (TAP) which included all forms of phosphorus, except...: HOW DOES IT VARY with existing concept or methodology?	Thank you for your comment. The current definition of available phosphorus often concentrate on dissolved phosphorus, as it is directly related to the growth condition of the plant and the effectiveness of fertilization. These definitions are made for earth soil, especially for agriculture or horticulture, in which the content of microbial biomass phosphorus is very small. However, if we ignore the microbial biomass phosphorus in our experiment, which is as much (or even twice) as dissolved phosphorus, we may draw the false conclusion that the PSBs could not dissolve phosphorus from the regolith, or falsely conclude from the rise of dissolved organic phosphorus of B. subtilis and B. licheniformis treatments that they could leach phosphorus from the simulant. In fact, the death of the cells of B. subtilis and B. licheniformis	No particular change has been made for this comment.

		led to the microbial biomass phosphorus releasing to the medium, causing the dissolved organic phosphorus to rise. We had discussed this in the manuscript.	
22	In " the sum of initial exogenous phosphorus input and all phosphorus dissociated by PSBs. : This is not clear? What was that exogenous P, in what form, etc.	Thank you for your comment. As we have described before, in No.15 of table 4, the exogenous phosphorus input contains dissolved phosphorus and microbial biomass phosphorus from the medium and the inoculant. To make our expressions clear, we have added such descriptions to the manuscript.	"(dissolved phosphorus and microbial biomass phosphorus from the medium and the inoculant)" is added next to the expression "exogenous phosphorus input" .
23	In " that there was little dissolved phosphorus in the lunar regolith simulant before the culture,: That was one of the issues I am still highlighting.	Thank you for your comment and the response in No.2, table 1 is also valid for this comment.	
24	In " entirely provided by liquid medium..": Weren't you able to provide liquid medium with no P available? How does the inoculant solution provide P before culture?	Thank you for your comment, but it is unrealistic to provide medium without phosphorus to PSBs. Yes, the PSBs that could dissolve phosphorus from the regolith may be able to grow in such medium, but other PSBs, like B. subtilis and B. licheniformis , is unable to grow in the medium without phosphorus. We could not know which PSB could leach phosphorus from the regolith before the experiment, so we believe that the best way to examine whether they can or can not dissociate phosphorus from the regolith is to allow them to grow in a ideal environment and measure the dynamics of the phosphorus content in different forms.	No particular change has been made for this comment.
25	In " This showed that B. mucilaginosus had a strong ability to excrete organic matter that .." : WAS this measured?	We are sorry for our improper expressions. Actually this is not measured in our experiment, and we could only make such assumptions according to the change of dissolved organic phosphorus content. To avoid confusion, we have adjusted the wording to make it more euphemistic, and show the fact that this is only our assumption.	"This showed that B. mucilaginosus had a strong ability to excrete organic matter that..." is replaced by "This indicated that B. mucilaginosus may have a strong ability to excrete organic matter that contains phosphorus compounds, which is worth for further investigation".
26	In ..."the contribution of adsorbed phosphorus to...": Did authors mean the adsorbed P (insoluble) in bacterial Cells?	Thank you for your comment, but we did not mean the insoluble adsorbed P in cells. As we had written in the definition, which is now moved to the section 2 of Supplementary Materials, adsorbed P means "the phosphorus adsorbed on the surface of regolith simulant particles through ligand adsorption or ion exchange adsorption".	No particular change has been made for this comment.
27	In section "Plant growth in the lunar surface regolith simulant treated with PSBs": L1 – 4 all are Methodology not results.	Thank you for your comment, but we have completely re-written the section of "Plant growth in the lunar surface regolith simulant treated with PSBs" after we have conducted the new supplementary cultivation experiment on Nicotiana benthamiana . We tried our best not to make similar mistakes that you and other reviewers have pointed out, and we managed to write our findings as clear as possible.	No particular change has been made for this comment.
28	In "Neither treatment hadgrowth.: THE sentence is not complete		
29	In ". The plants showed severeat 21 DAG. / this NEEDS TO BE EXPLAINED:rewritten		
30	In " The results suggested that the ...the simulant. : how,?		
31	In "...The higher phosphorus content indicated ... phosphorus elements. : Why?		
32	In "...which led to the increase of soluble salt content in the soil, resulting in severe salt stress on plants and inhibiting the further growth. : Do you think % soluble salt around 3-4% after 250 h of culture will inhibit plant growth? This is actually what authors measured in vitro, but should this be extrapolated to the in plant experiment as an explanation to the growth inhibition?	Thank you for your comment. We have examined the dynamics of soluble salt content in different treatments during and before the cultivation in our supplementary experiment. According to our results, The highest soluble salt content is just 1.18%(pre-cultured for 18 days, 24 DAS), much lower than that in the shaking flask experiment. However, we are also glad to find out that the inhibitory impact that 15 DAI treatment had on plant growth disappeared in our new cultivation experiment, after we adjusted our preparation method of cultivation substrate. We assumed that the high P level and relatively low salt content (compared with former 15 DAI treatment group) may contributed to the better growth of plants.	No particular change has been made for this comment.
33	In " Meanwhile, the experiment revealed the negative impacts caused by the prolonged treatment of PSBs, which may be caused by high soluble salt content.: THIS is still a speculative statement given that there is no evidence provided here to support quantitatively this conclusion.	The response above is also valid for this comment.	No particular change has been made for this comment.
34	In "It should be noted that the culture medium for B. mucilaginosus was silicate bacteria medium (glucose 5 g/L, MgSO4 0.5g /L, CaSO4·2H2O 0.1 g/L, Na2HPO4 2 g/L, FeCl3 0.005 g/L, pH=7.0).: authors needs to justify why?	Thank you for your comments. We use this medium to culture B. mucilaginosus because the provider suggest that it would have maximum growth in silicate bacteria medium. We have also added such descriptions in this section to clarify this issue.	The following statement is added to the section 2 of Methods: "as the provider of B. mucilaginosus suggest that it would have maximum growth in silicate bacteria medium."
35	In provided by our laboratory.: Please define the laboratory name	Thank you for your notice, and we have added the name of laboratory that provided the seeds.	The following statement is added to the section 3 of Methods: " The seeds used in the cultivation experiments were provided by professor Ronghui Pan, Zhejiang university, PRC. The seeds were collected in June, 2022."
36	In: The culture medium was sampled before and after culture, and..." was this AFTER INOCULATION?	Thank you for your comment. The sample was taken before the culture but after the inoculation, and we added the description to our manuscript.	"The culture medium was sampled before and after culture" is replaced by "The culture medium was sampled right after the inoculation, and after the culture process."
37	In: shaking flask culture experiment of PSBs with lunar regolith simulant: pleas improve the title	Thank you for your comment on the title of section 5, Methods. We have improved the title according to your comment, making it simple and informative.	The title of section 5, Methods, has been replaced with "Shaking flask experiment of PSBs"

38	In: "...and the possibility of microbial dissociation on insoluble inorganic phosphorus by PSBs...": here it is not clear what the author wanted to explain, as it is the statement is similar to the most literature about PSB aiming to solubilize insoluble P. Is the purpose here to solubilize P FROM REGOLITH, or add tricalcium P mixed it with regolith and quantify THE SOLUBLE ?	We partially agree with this comment. We realize that the meaning expressed by this sentence is somewhat repetitive and is not necessary for the context. Thus we have deleted it from our manuscript. The response on the comments about TCP could be found in No.2, table 1.	"To explore two problems.....inorganic phosphorus by PSBs" is removed from the section 4, Methods.
39	In: "The solid phase consisted of 40 g of the lunar surface regolith simulant which was pre-dried to constant weight at 70 °C in a 150 ml flask..." : GRANULOMETRIE?	We are sorry that we could not understand the word "GRANULOMETRIE" in this comment. According to our understanding, you may want us to describe the particle size of the regolith used for the culture. Therefore, we added the particle size of the regolith to the manuscript.	"lunar surface regolith simulant" in section 5, Methods has been replaced by "lunar surface regolith simulant (0.3-0.5 mm)" .
40	In: "...40g of solid phase within an ultra-clean workbench, and sealed a sealing film of air-permeability. Before inoculation, the mixed culture medium was cultured at 30°C and 180 rpm for 24 hours to check whether it was contaminated. : Authors needs to explain using 40g of simulant, and why not testing different amount before choosing only one for the rest of experiments (here 40g)?	Thank you for your comment. As a matter of fact, We have conducted a series of pre- experiments using different amount of regolith. As a similar comment has been asked by reviewer #1, we would like to guide you to the No.5, table 2 for detailed explanation.	No particular change has been made for this comment.
41	In : 6. Sampling : TITLE NEEDS TO BE INFORMATIVE	Thank you for your comment, and we have renamed this section.	Section 6 of Methods has been renamed as "Sampling method of the shaking flask experiment".
42	In: "The seeds used in the experiment included lettuce (Lactuca sativa) and common wheat (Triticum aestivum). According to the shaking flask culture experiment results, the tested PSBs were determined to be B. mucilaginosus , B. megaterium , and P. fluorescens . : Section 7 lacks explanation about the purpose of the seed germination experiment	We agree with your comment. We realized that the germination experiment is unnecessary in the main text, because it serves better as a pre-experiment of cultivation experiment than as a part of the main text. Therefore, we deleted the related descriptions, discussion and figures from the manuscript.	
43	In: The culture medium of the above three PSBs in the regolith simulant culture experiment at 0 and 21 DAI: not clear why 0 and 21?	Thank you for asking for clarification. In fact, we would like to find out whether there was any difference in the germination rate of seeds treated with the culture from before and after the culturing process. Frankly speaking, the name of "0 DAI" and "21 DAI" is not good, as 0 DAI stands for "before the culture", and 21 DAI stands for "after the culture". However, as we have deleted the discussion about the germination test from our manuscript, these parts are no longer needed to be adjusted.	No particular change has been made for this comment.
44	In: "was fully moistened with diluted culture medium or sterile water.: Thea reason why inoculation was done 0 & 21 DAI and using diluted PSB-rigolith culture is not clear In addition, it is questionable why Arabidopsis seeds were not tested for germination potential and rate along with wheat and lettuce, particularly Arabidopsis stands as the main tested plant species in this research.	Thank you for your comment. The reason why we used wheat and lettuce seeds for germination test was simple: we may actually grow these plants on the moon in the future. However, when doing cultivation experiments, it would need at least 50 g of the regolith simulant for each replicate in every treatment groups to grow lettuce, and even more to grow wheat, because of the relatively large weight and rich storage of nutrients of a single lettuce or wheat seed compared with Arabidopsis or tobacco. We have insufficient regolith to grow these plants. On the other hand, these "big" seeds may survive and grow normally for a relatively long period of time in a regolith with very poor fertility, with the supply nutrients from their inner storage. Therefore, wheat and lettuce seeds are less sensitive to the environment compared with Arabidopsis and tobacco. These are the reasons why we used Arabidopsis thaliana (and Nicotiana benthamiana for supplementary experiment) in our cultivation experiment.	No particular change has been made for this comment.
45	In: ". To stop culture, five bottles...": WHAT DOES IT MEAN BY "TO STOP CULTURE" and what is the purpose of this? My question here is, why arabidopsis growth was not tested under simultaneous inoculation and addition of simulant (this would have taken as a treatment), instead of adding a simulant that was previously inoculated and incubated then used to grow Arabidopsis ?	Thank you for asking for clarification. The original purpose of "to stop culture", was to kill PSBs from the regolith and stop the culture process. We later realized that this may not be necessary, as we assumed that living PSB cells could continue to dissociate phosphorus from the regolith during the cultivation period. Thus we had deleted this step, and the PSBs were remained alive during the cultivation. You may be satisfied about the "Not pre-cultured" treatment in our supplementary experiment, as the plants in this treatment were tested under simultaneous inoculation and addition of simulant.	"To stop culture" is deleted from section 7, Methods.
46	In: "...0.2 ml 1/4 MS medium: WHAT IS MS medium?	We are sorry for our wording. We have replaced "MS medium" with its full name, "Murashige & Skoog medium".	Changes have been made according to the left.
47	In: "However, due to pollution or other factors, some replicates were stopped from culture or cultivation during the culture.: In this case, how is it likely possible to statistically interpret results of treatments with less than 3 replicates according to the Supplementary Table 1.	Thank you for your comment, and we agree with you. As a critical comment listed in the table 1, we have placed the response to this comment in the row No.3 of table 1.	Changes have been made as described in row No.3, table1.

REVIEWERS' COMMENTS:

Reviewer #1 (Remarks to the Author):

The main idea of this work is that phosphate-solubilizing bacteria help to solubilize the phosphorus in the lunar regolith and improve plant growth. In order to prove their point, authors chose five different bacteria and tested their phosphate solubilizing capacity in cultures in which the lunar regolith was added. Later, they attempted to use a combination of three different bacteria in the lunar regolith stimulant; and assess the impact of the presence of bacteria to plant health. This is the second time I have reviewed this work. The authors made some significant changes since the first version of the manuscript.

- One of the main concerns about the first version of the manuscript was the number of replicates. The authors did not have enough replicates to claim the plant growth effects they suggested. In the first version of the manuscript, they used *Arabidopsis* as the plant model. Here, they changed it with *Nicotiana benthamiana* and they have an appropriate experimental design and sufficient replication to show the effects of bacteria on plant growth on lunar regolith.

- Another important change that was made is that the authors added the elemental analysis of the CAS9 lunar stimulant to the manuscript, which was missing in the first version. With this data, we can now see that this regolith stimulant has a similar elemental content to Apollo 13 samples.

- Authors improved the presentation of data as suggested.

Overall, the manuscript has seriously improved since the first version. They addressed my main concerns as well as Reviewer2's main concerns. My specific comments can be found below.

Table 2:

Point 1: Because the authors changed the plant model and have a significant effect of bacteria on plant performance (Fig 5 b and c), this comment is acceptable.

Point 2: I understand the budget issues that the authors had. But I think adding the soil extract + bacterial growth data on Figure 5 does not really prove the point because the soil is very different than lunar regolith. I would suggest that the Authors can move this data to a supplementary figure since the soil was not used in other parts of the manuscript. For Figure 5, please indicate clearly on the figure legend that figures 5 b and c is from the plants that are grown in regolith, not soil. Having the regolith-grown plants + soil pH data in the same figure makes things confusing.

Point 3: The authors changed the plant material used in the experiments, so no further comment on this one.

Point 5: Please add the relevant information described here to the manuscript. I think it is important to explicitly explain the optimization of these experiments to the audience since the lunar regolith experiments are still under a lot of optimization.

Point 8: I accept the results from the new cultivation experiment and agree with the authors.

Reviewer2 comments/ Table 3:

Point 3: Given that there is no oxidation experiment in the manuscript, I would agree with the authors that it is not necessary to add this to the discussion.

Reviewer #3 (Remarks to the Author):

Based on Authors' responses provided in detail in the R/A letter, I can say that I am satisfied with the

answers to all my comments. however, While scrolling the R1 version, it appears that authors need to carefully edit the Ms with some examples below:

In fig. 1: division of diviation (please check in the other figure captions)

Authors used the term "attached" in attached microbial biomass P: what is the reason of using this word, isn't simply to use microbial biomass P instead?

Are author sure the MBP is attahed on the surfac eof microbes, that's why the term is used?

It is advised avoiding personal pronouns, such as in the following example ("we" used) :

We then examined the properties of the lunar regolith simulant samples at different times during the culture process,

to confirm that the PSBs promoted the plant growth. Similar to the flask shaking experiment, we measured the pH and

OD600 of the soil extract, the available phosphorus content and the soluble salt content of the simulant(fig. 5d-g).

The pages are not umbred, which makes tough locating the suggested revisions

What did the authors want to mean by "which may be explained later"? this is not understood

In "In conclusion, our study shows that three PSBs, including..." I think "including" should be deleted

In "The samples were then digested and the composition...": what are these elements?

Please note these comments are only exmaples, and authors are highly recommended to check the entire Ms.

Dear reviewers,

We deeply appreciate your general approval of our manuscript, and we also attach great importance to your comments. Here, we have listed the point-to-point responses to your comments in the tables, and we also adjusted minor mistakes (e.g. language, numbers, type-setting) in our manuscript, which you could find in the marked-up version of the manuscript.

Again, we would like to thank you with great gratitude, and we wish you every success.

If there is any questions, please contact us.

Authors

The order of the tables:

Table 1, Comments from reviewer #1;

Table 2, Comments from reviewer #3.

Table 1. Comments from reviewer #1;

No.	Comments from reviewer #1	Author respond	Change made
1	I understand the budget issues that the authors had. But I think adding the soil extract + bacterial growth data on Figure 5 does not really prove the point because the soil is very different than lunar regolith. I would suggest that the Authors can move this data to a supplementary figure since the soil was not used in other parts of the manuscript. For Figure 5, please indicate clearly on the figure legend that figures 5 b and c is from the plants that are grown in regolith, not soil. Having the regolith-grown plants + soil pH data in the same figure makes things confusing.	We are sorry that the wording in the manuscript caused confusion. As a matter of fact, we used the expression "soil pH" and "soil extract" because it is considered as a convention in the soil science. They described the status of the liquid extract from the regolith treated in the cultivation experiment. To avoid further confusion, we changed the word "soil" to "regolith", as we have used in other parts of the manuscript.	The expression "soil extract" and "soil ph" that describe the regolith extract is replaced by "regolith extract" and "pH of the regolith extract" from the manuscript and the Fig. 6.
2	Please add the relevant information described here to the manuscript. I think it is important to explicitly explain the optimization of these experiments to the audience since the lunar regolith experiments are still under a lot of optimization.	We have added relevant information that describes the optimization about regolith experiment to Methods, Shaking flask experiment of PSBs. However, we could not sure if this is the information that you mentioned, because the comment that we receive is incomplete. We hope this could meet your satisfaction. In addition, we have already discussed the possible method of experimental optimization, for example, combining with soil-like substrate experiments.	Relevant information that describes the optimization about regolith experiment is added to Methods, Shaking flask experiment of PSBs.

Table 2. Comments from reviewer #3.

No.	Comments from reviewer #3	Author respond	Change made
1	In fig. 1: division of deviation (please check in the other figure captions)	We are sorry for the misspelling. We found out that the correct spelling is "Deviation", and we have corrected them in the manuscript. We have also corrected other misspelled words in the manuscript, for example, "hoticulture", "rosseta", "droped", "attatched" are replaced by "horticulture", "rosette", "dropped", "attached".	Changes have been made as described on the left.
2	Authors used the term "attached" in attached microbial biomass P: what is the reason of using this word, isn't simply to use microbial biomass P instead? Are author sure the MBP is attached on the surface of microbes, that's why the term is used?	We are sorry for the misunderstanding. According to the definition, attached microbial biomass P means the P that contains in the PSB cells attached to the regolith particles. We did the clarification in the first version of our manuscript, but it was later deleted as being considered as a part of methodology. Here we put it back to the manuscript, and we keep the length of description minimum. We did not distinguish the intracellular P and	The supplementary information is added to the place where "Attached microbial biomass phosphorus" first appear in the manuscript.

		the P that attached to the membrane in our study.	
3	It is advised avoiding personal pronouns, such as in the following example ("we" used) : We then examined the properties of the lunar regolith simulant samples at different times during the culture process, to confirm that the PSBs promoted the plant growth. Similar to the flask shaking experiment, we measured the pH and OD600 of the soil extract, the available phosphorus content and the soluble salt content of the simulant(fig. 5d-g). The pages are not umbred, which makes tough locating the suggested revisions	Thank you for your comment and we have realized the need of using a passive voice. We have examined the manuscript and rewrote the sentences that required the passive voice.	Please refer to the manuscript. Too many changes have been made according to this comment, not suitable for presentation in a table with limited space.
4	What did the authors want to mean by "which may be explained later"? this is not understood	We are sorry for the vague expression. We intended to say that we may explain this in later studies, but not in this manuscript. Therefore we have changed the expression.	"which may be explained later" is replaced by "which may be explained in further studies".
5	In "In conclusion, our study shows that three PSBs, including..." I think "including" should be deleted	We agree with your comment and we have deleted "three PSBs, including".	Changes have been made as described on the left.
6	In "The samples were then digested and the composition...": what are these elements?	We again apologize for the incomplete description.	"The composition of some elements" is replaced by "The mass ratio of Fe, Ca, Mg, Na, K, P, and Mn"